# Evaluation and Comparison of Spatial Clustering for Solar Irradiance Time Series

Luis Garcia-Gutierrez [1,2], Cyril Voyant [1,3], Gilles Notton [1,*] and Javier Almorox [4]

1   Sciences for Environment Laboratory, University of Corsica Pasquale Paoli, UMR CNRS 6134, 20000 Ajaccio, France
2   Department of Electronic Engineering, School of Exact Sciences and Engineering, Universidad Sergio Arboleda, 11000 Bogotá, Colombia
3   Radiotherapy Unit, Hospital of Castellucio, 20000 Ajaccio, France
4   Universidad Politécnica de Madrid, UPM, Avd. Puerta de Hierro, 2, 28040 Madrid, Spain
*   Correspondence: notton_g@univ-corse.fr

**Abstract:** This work exposes an innovative clustering method of solar radiation stations, using static and dynamic parameters, based on multi-criteria analysis for future objectives to make the forecasting of the solar resource easier. The innovation relies on a characterization of solar irradiation from both a quantitative point of view and a qualitative one (variability of the intermittent sources). Each of the 76 Spanish stations studied is firstly characterized by static parameters of solar radiation distributions (mean, standard deviation, skewness, and kurtosis) and then by dynamic ones (Hurst exponent and forecastability coefficient, which is a new concept to characterize the "difficulty" to predict the solar radiation intermittence) that are rarely used, or even never used previously, in such a study. A redundancy analysis shows that, among all the explanatory variables used, three are essential and sufficient to characterize the solar irradiation behavior of each site; thus, in accordance with the principle of parsimony, only the mean and the two dynamic parameters are used. Four clustering methods were applied to identify geographical areas with similar solar irradiation characteristics at a half-an-hour time step: hierarchical, k-means, k-medoids, and spectral cluster. The achieved clusters are compared with each other and with an updated Köppen–Geiger climate classification. The relationship between clusters is analyzed according to the Rand and Jaccard Indexes. For both cases (five and three classes), the hierarchical clustering algorithm is the closest to the Köppen classification. An evaluation of the clustering algorithms' performance shows no interest in implementing k-means and spectral clustering simultaneously since the results are similar by more than 90% for three and five classes. The recommendations for operating a solar radiation clustering are to use k-means or hierarchical clustering based on mean, Hurst exponent, and forecastability parameters.

**Keywords:** solar irradiation; data mining; time-series clustering; artificial intelligence; statistics methods

## 1. Introduction

Renewable energy is commonly referred to as clean energy, as it comes from natural processes or sources that are constantly being renewed. However, it is recognized that the knowledge of the "availability" and "variability" of these renewable sources [1] is an important task and a great challenge, considering that the development of solar energy over the world is increasing and will continue to increase in the future [2–4].

The main problem of this energy source, which could inhibit or reduce its development, is its intermittence and randomness, which make its management difficult for an energy system operator [5]. Thus, to reduce and even to delete, at least partially, this inconvenience, three research axes must be developed in parallel:

- The development of energy storages: They will allow us to store the energy in excess and to restore it when the load requires it [6];

- The development of smart concepts of electrical grid: The produced and consumed energy and power must be managed efficiently with smart algorithms, using information and communication technologies [7];
- The forecasting of the intermittent renewable energy production: Solar- or wind-prediction tools must be integrated into energy and power management systems to anticipate the future actions [8].

The benefit of solar radiation prediction is effectively proven [9]. There are a large number of publications in the literature that address the problem of solar-irradiance prediction. The techniques applied to prediction depend on the time horizon of the prediction and the time step of the solar irradiation [10]. Between these methods, time-series-based methods are based on statistical models applied to ground data measured in the past and at the instant of the prediction, using regression techniques [11], artificial intelligence methods [12], or statistical approaches [13]. They are used to forecast solar data from some minutes to about a 6 h horizon. They often have the disadvantage of being efficient only for the site on which the model has been trained and to be not generalizable elsewhere.

It should be interesting to verify if a model developed on a given site must be applied to other ones with "similar" characteristics not only from the received-solar-energy point of view but also from the point of view of the dynamic behavior of the solar irradiation one. With this final objective, the purpose of this study was to group the meteorological stations by "solar radiation affinities".

Clustering is a technique used in data mining to identify clusters of elements according to a measure of similarity between them. In addition, clustering to minimize the dimensionality of the data while handling a large number of variables is used [14]. On the other hand, the clustering of geographically spaced data has been addressed in the literature. Still, this one seeks to cluster similar or geographically proximate data in geographic-information-system (GIS) information layers [15].

The study of the behavior of times series applied to a large territory by using statistic and artificial intelligence (AI) methods, grouping in "clusters" time series with the same behavior (i.e., with the same variability), was realized particularly in two papers written by Warren Liao [16] and Wang et al. [17]. The first one [16] summarized previous works on clustering time-series data into various application domains and discussed criteria for evaluating the performance of clustering results and measures for determining the similarity/dissimilarity between two time series. The second one [17] realized a high-dimensional time-series clustering work, using algorithms based on distance metrics that generally fail because they cannot handle missing data. Instead, time series are clustered based on global features extracted from each series and can be fed into clustering algorithms. In these works, statistical operations describe the time series: trend, seasonality, periodicity, serial correlation, kurtosis, chaos, nonlinearity, and self-similarity.

Reviews papers such as References [18,19] present states-of-the-art in the last decade that introduces some fundamental concepts of time-series analysis, classification, and clustering methods based on observations, time and frequency, and the similarity between series and dimensionality reduction.

Focusing now on the clustering in renewable energy sources, Tripathi et al. [20] described various AI techniques applied to different renewable energies, particularly photovoltaic ones. They realized a clustering of datasets for analyzing the most suitable location of solar plants according to the study of similar meteorological conditions.

Hartmann [21] compared various methods to categorize clear, cloudy, and partiality cloudy sky days, using a one-year solar-irradiance dataset from Budapest (Hungary). Six methods were compared deterministically and non-deterministically (k-means clustering, Fuzzy c-Means, and Multiple Fuzzy C-Means) with different temporal resolutions. The comparison aimed to reveal the strengths and weaknesses of the applied methods. Unfortunately, the results obtained with the implemented methods are limited, as they are susceptible to the input data, and the categorization is often inconsistent among them.

In Reference [22], the authors aimed to predict the global hourly solar radiation of the next day and used clustering. The k-means clustering algorithm was proposed to classify four kinds of days: clear, cloudy, cloudy in the morning, and cloudy in the afternoon. They were combined with regression algorithms (decision trees, support vector machines, and artificial neural networks) to estimate the day-ahead clearness index in Malaga, Spain.

Liu et al. [23] proposed a solar irradiance classification of 98 stations with solar irradiance sensors and 562 stations without irradiance data. A k-means and a support-vector-machine genetic algorithm were used to perform this clustering. Models were developed by introducing geographical parameters, including latitude and altitude. Hassan et al. [24] sought to predict hourly global irradiance profiles from daily global irradiance records from six sites in the North African Sahara. They started from a prior categorization of the hourly observations by using a k-means clustering algorithm, followed by a nonparametric approximation of the function, using the multilayer perceptron artificial neural network.

Laguarda et al. [25] studied the relationship between the solar resource and the climatology in Uruguay (condition of the phenomenon called Niño or Niña). The input information consisted of global-horizontal-irradiance (GHI) estimations from daily satellite images and regionalization of the meteorological stations based on the geographical position of the station. A Principal Component Analysis (PCA) was used to decrease the dimensionality of the series. The clustering algorithm, based on the changes in the station for different years, was a k-means/ward.

Pham et al. [26] applied a k-means clustering to satellite-based daily global horizontal irradiation for spatial-variability analysis and regionalization in different regions of Vietnam.

Maldona-Salgero et al. [27] proposed a methodology to characterize and cluster the spatiotemporal daily global horizontal solar-resource variability in the Spanish territory. They used a hierarchical clustering technique to classify the spatial data, and different time windows were subsequently evaluated. The parameters used in this study are averaged (yearly, seasonal, and monthly) values of daily horizontal global irradiances.

An unsupervised clustering-based (UC-based) solar forecasting method was developed by Feng et al. [28] for short-term (1-h-ahead) global horizontal irradiance forecasting. The daily GHI time series is clustered by an Optimized Cross-Validated Clustering (OC-CUR) method.

Malakar et al. [29] proposed a novel short-term (2-h-ahead) solar forecasting approach that uses a clustering (k-medoid clustering method) on the basis of cloud parameters as a preprocessing step. To ensure broader variation in cloud movements, neighboring stations were used that were selected by using a dynamic time warping (DTW)-based similarity score. The method is very interesting, although complex. The proposed model achieved 19.74% less nRMSE compared to the benchmarks.

The majority of the papers presented above propose methods and modes of validation crucial for the clustering of solar radiation; however, they are limited, for the most part, to a single-criterion study most often relating to a geographical criterion and daily and monthly data (time step not really interesting in practice for solar applications). In order to address these limitations, we propose to develop a new methodology, particularly by studying non-geographical parameters and statistical parameters revealing the dynamics of each hourly or intra-hourly clear-sky index time series. Various techniques of clustering [30] are applied to group the meteorological stations by "solar radiation affinities", i.e., to group by similarity or proximity criteria according to the static and dynamic characteristics of the time series. Clustering is not an exact science; totally contradictory results can be linked to the use of poor-quality measures. It is therefore essential, in the field of solar engineering, in addition to stating this characteristic, to identify, quantify, and correct possible problems. The various steps of this work are as follows (rarely are all of these steps studied in the same paper):

- To evaluate the quality of the data provided by the meteorological stations because the measure of solar irradiation is often accompanied by errors due to pyranometer calibration, surrounding effects, or data-acquisition-system failure;

- To determine the best descriptors of static (e.g., descriptive statistics measures) and dynamic (e.g., Hurst exponent (H), forecastability coefficient (F), etc.) characteristics of the series and to determine the redundancy of these descriptors;
- To implement clustering algorithms such as k-mean, k-medoids, spectral, and hierarchical clustering;
- To compare the clusters obtained by each method with each other and the well-known Köppen classification, characterizing the areas according to climate, solar irradiation, precipitation, and temperatures.

The paper continues as follows: Section 2 implements tools that analyze the accuracy and quality of the measured data; Section 3 evaluates the best criteria to characterize solar time series; Section 4 presents the Köppen climatic classification and the clustering methods that are implemented on the solar irradiance time series; Section 5 implements some clustering algorithms and compares the clustering results; at last, in Sections 6 and 7, the perspectives of the work are described and conclusions are drawn, respectively.

The major contributions of the paper are as follows:

- The objective of the clustering is to improve the efficiency of the forecasting of the solar radiation; the objective was found only in one reference [29];
- The utilization of several parameters to characterize each site;
- The utilization of dynamic parameters to characterize each meteorological station: Hurst exponent already used for other meteorological time series but not for solar irradiances, and mainly the forecastability coefficient, which is a new concept—the utilization of several clustering methods and their comparison between them and with the well-known Köppen classification (upgraded for Spain).

## 2. Solar Irradiation Time-Series Analysis

To conduct this study, large databases (with an intra-hour time step) collected on several meteorological stations were needed. The solar irradiances for the present work were provided by the "Sistema de Información Agroclimática para el Regadío" (SIAR) system [31], consisting of 448 meteorological stations (https://eportal.mapa.gob.es//websiar/Inicio.aspx) (accessed on 2 June 2021). The data are free but are not open-access. These stations are concentrated in areas suitable for agriculture, with low slopes and measure temperature, rainfall, horizontal solar radiation, humidity, wind speed, and direction. Each weather station undergoes maintenance every six months and calibration every year. Note that rainfall, humidity, and solar irradiance measurements are used to determine weather stations in the Köppen [32] class.

### 2.1. Data Measures and Stations

The Spanish Ministry of Agriculture, Fisheries, and Food [31] make available an Agroclimatic Information System for Irrigation (SIAR) with measured data recorded every 30 min on 542 sites (only 448 are active; https://eportal.mapa.gob.es//websiar/Inicio.aspx, accessed on 2 June 2021). The position of the meteorological stations used in this work is shown in Figure 1. Solar irradiance measurements obtained by pyranometers are verified according to the International Standard ISO 9847 Solar energy Calibration. For the purposes of this study, global horizontal radiation time series (GHI) from 1 January 2017 to 31 December 2020, with a 30 min time step, were used. It is obvious that a reliable measure of meteorological data is not always an easy task, and even the maintenance of the station is well realized; quality control must be applied, mainly on solar radiation data.

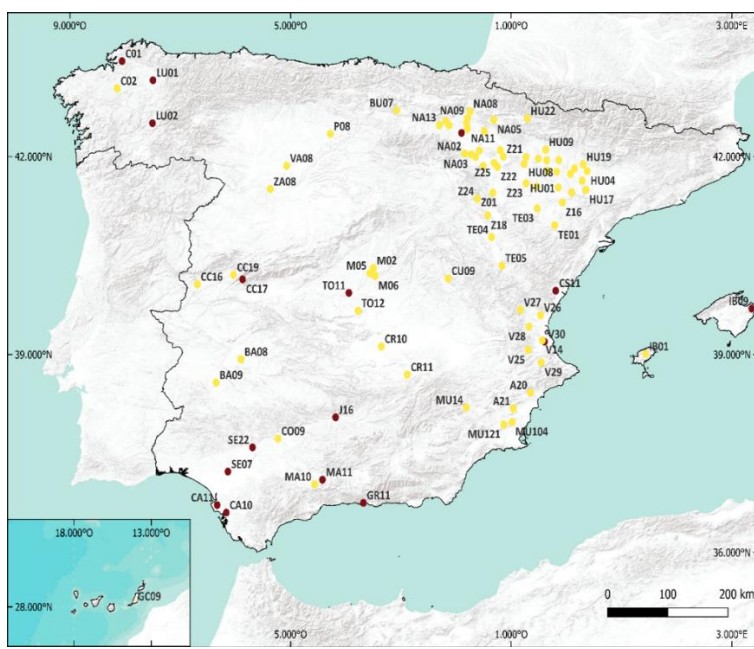

**Figure 1.** Positions of the weather stations considered in this study with solar data available from 1 January 2017 to 31 December 2020. In red, stations are not used because the data are not considered reliable. In yellow, the stations used. The geographical coordinates of the weather stations can be found in Reference [31].

Before explaining the quality-control method (Section 2.3), the clear-sky solar radiation model (Section 2.2) is briefly described here.

### 2.2. Clear Sky Model

The solar irradiation time-series GHI (for global horizontal irradiation or irradiance) exhibits a seasonality and a periodicity (a yearly cycle and a diurnal one). Most of the machine learning and AI methods can only be used with stationary time series [33] (the "weak" assumption is sufficient). To "delete" this periodicity, it is common to compute a ratio called clear-sky index and defined as GHI on the solar irradiance in clear-sky conditions (denoted GHIcs), kt = GHI/GHIcs. This one results in a normalized quantity theoretically comprised between 0 and 1. A clear-sky model estimates the solar irradiance that reaches the ground surface when the sky is clear without clouds. Such models were developed and differ from each other mainly in the inputs needed by each model [34]; they generally used meteorological variables (such as ozone layer thickness, precipitable water, aerosol optical depth, etc.) and solar geometry (solar elevation, declination, etc.) [35]. The most widely used clear-sky models are the Solis model developed by Mueller et al. [36] and simplified by Ineichen [37], the European Solar Radiation Atlas (ESRA) model [38], the Reference Evaluation on Solar Transmittance 2 (REST2) model [39], and the McClear Model [40–42] (this list is not exhaustive). The New Heliosat-4 method [40] is an operational tool for solar irradiance monitoring in the framework of the projects "Monitoring Atmospheric Composition and Climate (MACC)" [43] and the Copernicus program. This method processes Meteosat images to create the CAMs radiation service (Copernicus Atmosphere Monitoring Service) used to calculate the clear-sky solar radiation, to estimate the cloud effect (coupled with APOLLO/SEV from DLR), and the ground albedo (derived from MODIS).

It allows us to have an efficient model with a fast implementation because it is available via a web service that gives time series of horizontal global, horizontal diffuse, and normal beam irradiances for a given point and a given period, from 2004 up to current day − 2 (minus two days), with a time step from 1 min to 1 day. For this paper, only the data of clear-sky solar irradiance were uploaded via http://www.soda-pro.com (accessed on 10 March 2021). More information about the McClear Model and the upload platform can

be found in references [40,41]. Figure 2 shows, as an example, the superposition of the series generated with this model (clear sky) and SIAR measurements (in real meteorological conditions) on the same locations (here, stations V29 and C01; see Figure 1).

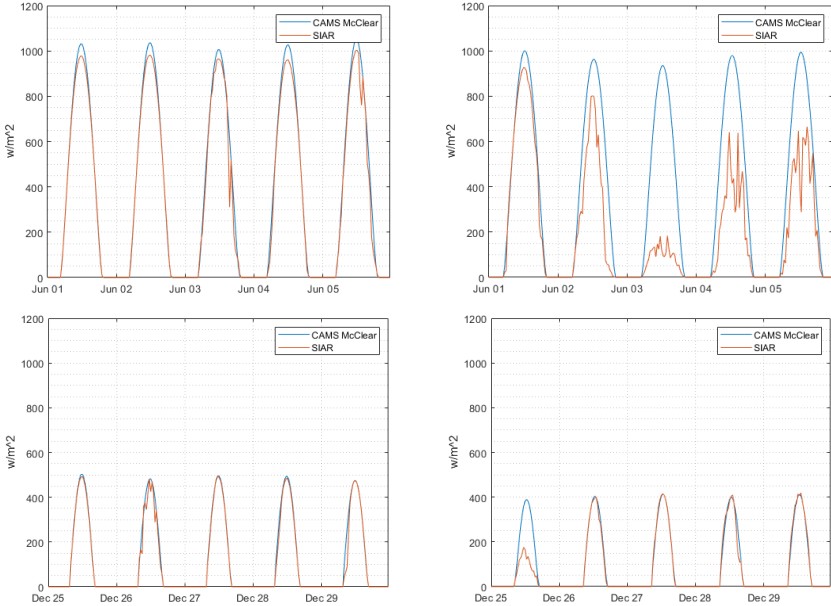

**Figure 2.** Presentation of GHI and GHIcs for 2 meteorological stations (V29 and C01) and 2 periods of 5 days in 2020 (winter and spring).

It is clear that the clear-sky irradiance estimated by CAMs can be quasi-perfectly superimposed onto the data measured on the ground level for days with clear skies. Thus, the quality of the CAM model is shown. It is possible to assess, more globally, the adequacy between measurement and model over a significant period (annual) for the same stations (see Figure 3).

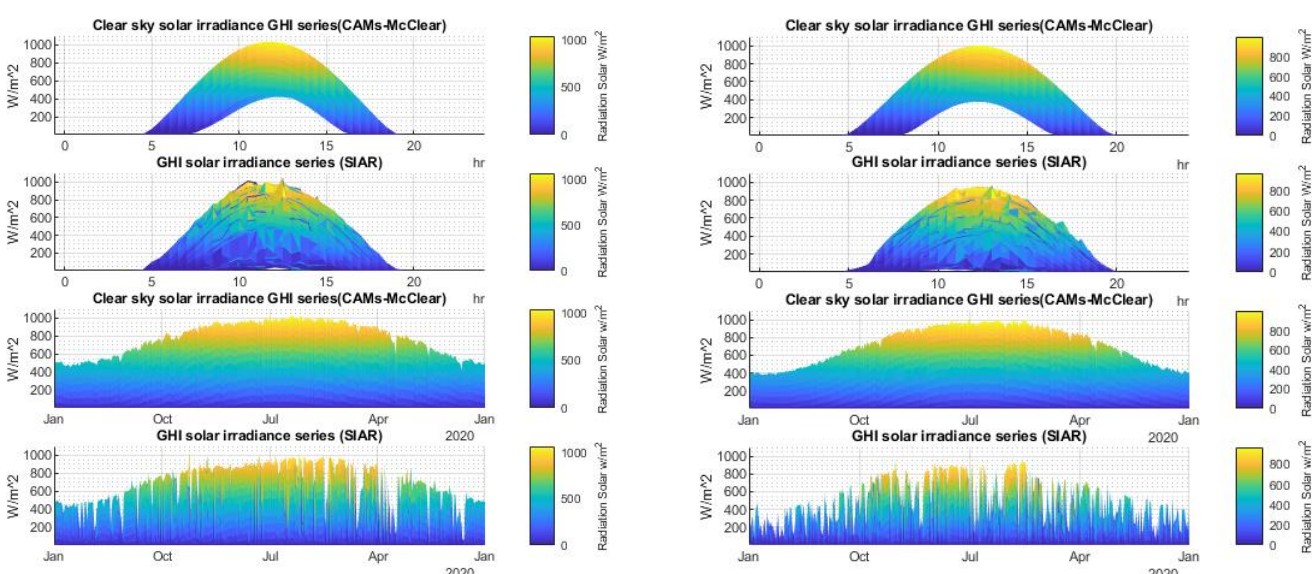

**Figure 3.** Presentation of GHI and GHIcs for 2 meteorological stations (V29 and C01) in 2020.

Using such a methodology, all the data, for each station, were carefully checked, and only stations having passed the tests correctly were selected in this study. As an intermediate outcome, the McClear model provides a good estimate of the clear-sky irradiance, but as shown in the next section, the results can still be improved.

### 2.3. Data Quality Control

A necessary condition to establish objective conclusions in clustering is to work with valid data. For this reason, the quality control of measured (SIAR) and estimated time series in clear-sky conditions (CAMs McClear) should be performed. Generally, several problems occur in solar-data acquisition, such as failures of the acquisition system, synchronization issues, incorrect measurements at sunset or sunrise (due to high azimuth angles), or other artifacts. Espinar et al. [44] and El Alani et al. [45] studied this critical topic of quality check and proposed efficient Quality-Control Procedures (QCPs) for solar data. They showed that, after checking the plausibility of data, some visual support (graphs and histograms) helps in the interpretation of this quality check. We drew inspiration from these two papers to establish our own QCP, which is described in Figure 4.

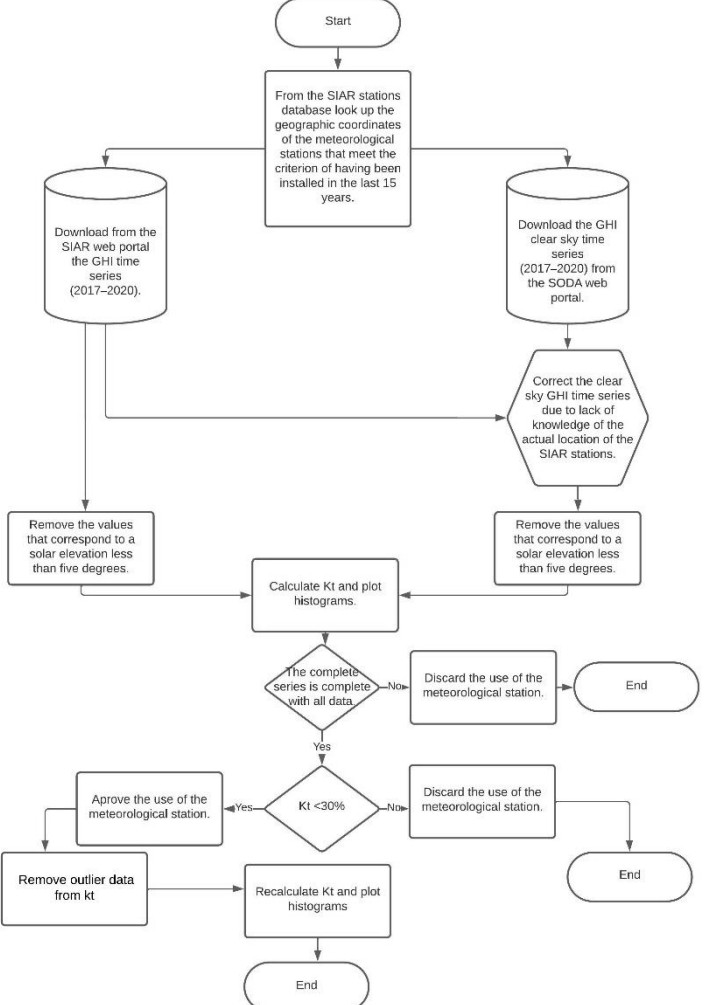

**Figure 4.** Proposed methodology for evaluation, correction, and cleaning of solar-irradiance data series.

We modified the previous methods a little bit in view to, in our opinion, make them more relevant for the data studied in this study: kt analysis based on geographic clustering. These processes are essential to extract relevant information for data mining and analysis tasks of solar-energy time series [46]. The QCP methodology is based on four main steps:

- GHIcs correction related to a possible offset corresponding to an erroneous time stamp: clock issue or stations' location approximation (see Section 2.3.1);
- Visual validation of kt series to easily identify significant measurement errors (see Section 2.3.2).
- Quality evaluation of data provided by SIAR system and discarding of the stations with more than 30% of over-irradiance data (see Section 2.3.2).

- Detection of the kt outlier. The data that are much larger than 1.2 are removed (see Section 2.3.3).

### 2.3.1. Time-Stamp Correction

The elimination or minimization of uncertainties in the kt time series must be realized [47–49], so the results will be relevant only if the clear sky and the measured solar irradiances are correctly time-stamped. It is quite common to observe a time lag (ranging from a few minutes to about 20 min) between the two types of series defining the clear-sky index: GHI and GHIcs. The effect is that it is systematically observed in higher measurements than the clear sky at sunset or sunrise. There are several explanations that can be put forward. The first is related to a mislocation (or lack of precision) of the measurement stations (which is unlikely but possible in some cases). Then an error can occur when entering the geographical characteristics of the stations in the McClear interface or a problem of clock between the measurement and the estimate by clear sky. A retiming strategy is proposed to minimize the delay between two time series. For this purpose, the methodology is as follows:

- Operate a clear-sky shift ($\Delta t$) from $-60$ min to $+60$ min (by steps of 1 min), based on a simple linear interpolation (121 time series are obtained).
- Compute for the 121 GHIcs series generated previously the mean square error (MSE) compared to measurement (GHI).
- Retain $\Delta t$, the offset that allows us to obtain the minimum MSE.
- Propose the new corrected clear-sky series, which corresponds to the linear interpolation of the series shifted by the offset, $\Delta t$ ($GHI_{cs}^{Corrected}(t) = GHI_{cs}(t + \Delta t)$).

The offset, $\Delta t$, is thus the parameter to find and then apply to GHIcs in order to find a better concordance in terms of the MSE between GHI and new GHIcs.

As an example, the impact of this shift is seen in Figure 5. Considering the correction and 1 year of acquisition (30 min time step), the average value of kt is decreased by almost 1%, from 0.5814 to 0.5761, and the variance by 4.2% (from 0.1136 to 0.1088).

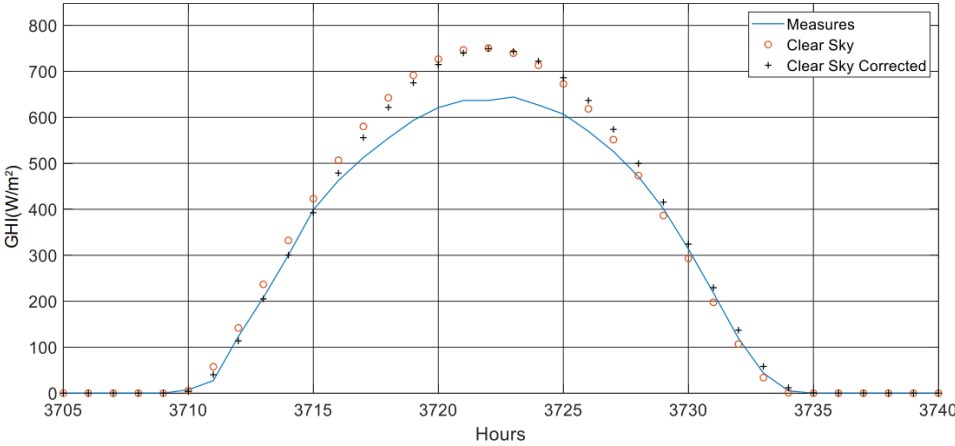

**Figure 5.** Clear-sky modeling corrected with 20 min offset in A Capela Station (C01) during 2020.

Now that the series is correctly recalibrated, we have to check if they are all usable.

### 2.3.2. Validation of the SIAR GHI Time Series

Even if the data series were verified by the SIAR's quality check, it seems that it is useful to realize a new checking by using the visual procedure proposed by Reference [45] and completed by some tests on the kt values. The solar irradiance data that are used here are the average horizontal global solar irradiance for 30 min on the period 1 January 2017 to 31 December 2020. For each station, the following steps are realized:

Visual validation: Measured data (SIAR) are plotted in a 2D graph (Figure 6); in the same graph are drawn the theoretical sunrise (red color) and sunset hour (green color). This

analysis allows for a visual identification of errors over time to visualize missing values, issues in time reference, and abnormal values. The two meteorological stations taken as an example passed the test, although an asymmetry is noted between the measurements above the green dotted line and below the red dotted line. Moreover, the measure begins before theoretical sunrise. This may indicate a little measurement error or sensor calibration error. This test allows, by its nature, to point out only the big measurement errors and is extremely time-consuming but necessary if one wants to use clean data. The other visual tests performed are not detailed here, but interested readers can refer to Reference [45].

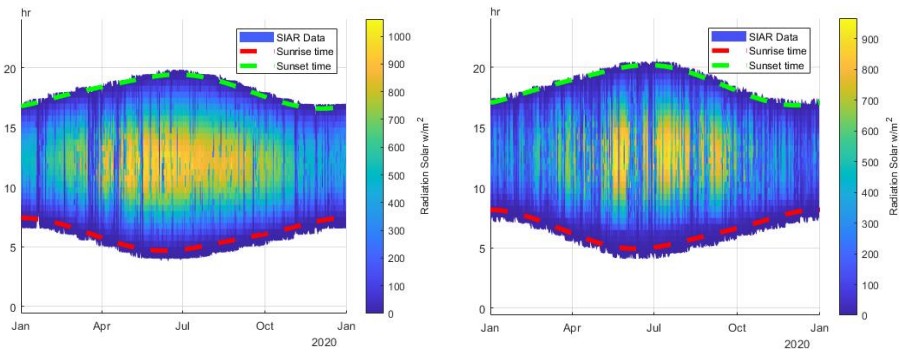

**Figure 6.** Visual validation for V29 and C01. *X*-axis is the day; *Y*-axis is the time in Universal Time Coordinated (UTC). The color of the surface, from blue (0 W·m$^{-2}$) to yellow (1000 W·m$^{-2}$), is the value of the data.

In what follows, a filtering of the time series is systematically performed to eliminate the night hours. GHI and GHIcs irradiance values are removed for the hours of the day related to a solar angle less than 5 degrees.

Over-irradiance test: This test is based on the exploitation of kt histograms, as seen in Figure 7. Although easily automated, it is interesting to look at the shape of the histogram. The experienced user will be able to easily point out measurement errors and stations to remove from the study. Another way is to quantify the number of occurrences where kt is greater than 1. We have arbitrarily chosen a limit of 30%. Thus, if for a series, there is more than 30% of over-irradiance data (which is rare or impossible for 30 min time granularity), we consider that it is preferable to not use the station.

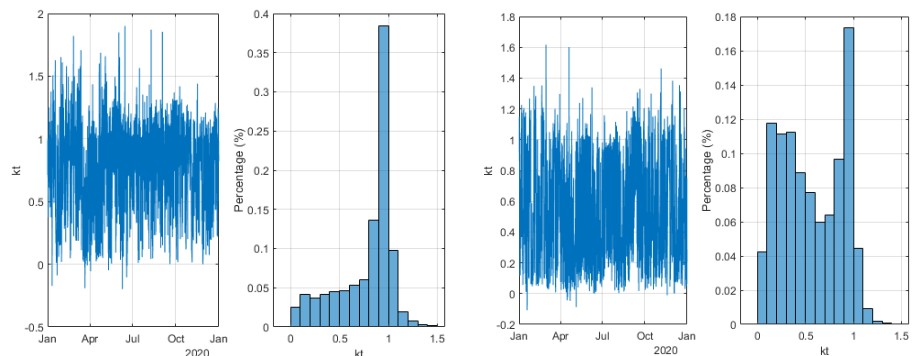

**Figure 7.** The kt profiles and histograms for V29 and C01.

### 2.3.3. Outlier Cleaning Methodology

Sometimes outlier values can pollute some interpretations, especially when a static treatment requires a normalization of the data. A maximum kt value equal to 2.5 or 1.2 does not induce the same effects. To counter this effect, kt values higher than 1.2 are removed. Even if some meteorological conditions can be met and induce over-irradiance, by experience with kt series with a time step of 30 min, these values are unlikely to be withdrawn to avoid a statistical bias in our treatments.

### 2.4. Results of the Data Quality Control

After applying the previously described process to the 94 stations, only 76 stations passed the tests (Figure 1). Table 1 shows the number of stations, total active, and validated by year.

**Table 1.** SIAR stations and quality control.

|  | **2020** | **2019** | **2018** | **2017** |
|---|---|---|---|---|
| Total SIAR station |  | 542 |  |  |
| Active SIAR Stations |  | 448 |  |  |
| Selected * SIAR Stations | 94 | 84 | 82 | 78 |
| Discarded ** SIAR stations | 10 | 2 | 4 | 2 |
| Subtotal | 84 | 82 | 78 | 76 |

* Solar data available and ** test failure (Section 2.3).

Although there are a lot of losses after completing the many tests presented above, there are still 76, which is rare in this kind of study. What is even more rare is to have simulations made with only corrected, validated, and filtered data.

## 3. Characterization of Solar Irradiance Time Series

Seventy-six GHI time series (one per location) measured between 1 January 2017 and 31 December 2020 were used. These observations were collected every 30 min. Each GHI series comprises 70,128 elements, for a total of 5,329,728 measurements. In addition, for each site and each 30 min, the GHIcs and kt (GHI/GHIcs) are computed; that is, more than 15 million data are created. As mentioned in the paper's Introduction, each site must be characterized from the available-solar-energy point of view, not only with the mean value or the cumulative GHI but mainly by the distribution or variation of the solar radiation. The notions of variability, forecastability, and dynamics of the series must be considered. The statistical parameters that are the most appropriate for this study are divided into two classes: the first one concerns static parameters (mean, standard deviation, skewness, and kurtosis), and the second one concerns the dynamic parameters (Hurst exponent (H) and forecastability coefficient (F)).

### 3.1. Static Parameters

A characterization of the data includes position, dispersion, symmetry, and feature of time-series distributions [50]. As performed in Reference [51], the mean is used to compare the different measures. It is the choice of simplicity and has thus left aside the other measures of central tendency (median and mode) and non-central tendency (quartiles, deciles, and percentiles). As it is important to consider the dispersion of the series, the standard deviation is used. The two other parameters are well-known in statistics:

- Skewness describes how much the statistical data distribution is asymmetrical from the normal distribution, where distribution is equally divided on each side [52–55]; it is defined by the following:

$$Skewness = \frac{\mu_3}{\sigma^3} \tag{1}$$

where $\mu_3$ is the third central moment and $\sigma$ is the the standard deviation.

- Kurtosis measures the "tailedness" of the probability distribution of a real-valued random variable [54–57] and is defined by the following equation:

$$Tailedness = \frac{\mu_4}{\sigma^4} \tag{2}$$

where $\mu_4$ is the fourth central moment.

### 3.2. Dynamic Parameters

A fundamental task in these statistical analyses is to characterize the location and variability of a dataset [58]. In order to analyze and compare the dynamic behavior of irradiance time series, the use of the Hurst exponent (H) and forecastability coefficient (F) is proposed. H is related to fractal geometry at different scales (concept defined by Mandelbrot [59]). The main idea states that fractals are objects that have a similar appearance when observed at different scales and have details that cannot be studied by Euclidean geometry. H was used in previous studies to characterize the variability of meteorological parameters [60–62]. The values of H for a time series with *n* components are in the range of 0 and 1 and are computed from the expected value ($E[x]$):

$$E\left[\frac{R(n)}{S(n)}\right] = \propto n^{\text{H}} \text{ as } n \to \infty \tag{3}$$

where $R(n)$ is the range of the *n* cumulative deviations and $S(n)$ is the sum of the first *n* standard deviations. According to the value of H, any time series can be classified into one of the three categories:

- If H < 0.5, the series is anti-persistent. The closer the value is to 0, the stronger the mean-reversion process is. In practice, it means that a high value is followed by a low value, and vice versa.
- If H = 0.5, the series is totally random.
- If H > 0.5, the series is persistent. The closer the value is to 1, the stronger the trend.

There are several methods to calculate H. We used rescaled range analysis, average wavelet coefficient, and periodogram regression detailed in References [63,64]. The different techniques are close in terms of results, with a difference of less than 10%. Therefore, we did not know which one to use, so we decided to average the results obtained with the three methods.

The *F* coefficient describes how a model trajectory diverges from a true system trajectory. It is bounded between 0% and 100% and provides insight into the extent to which solar radiation time series can be predicted. This parameter can be estimated from a Monte Carlo method to determine $RMSE_{max}$ (the maximum error related to a site where the GHI randomly oscillates between 0 and $GHI_{CS}$) and the RMSE of the smart persistence predictor ($RMSE_p$) [65]. *F* can be calculated with Equation (1):

$$F \approx 100\% \times \left(1 - RMSE_p \ / \ RMSE_{max}\right) \tag{4}$$

The results for all the stations are given in Appendix A Table A1. For more details concerning this metric, interested readers can refer to Reference [66].

### 3.3. Correlation between Statistical Parameters—Redundancy

This section assesses whether the parameters defined above are not redundant and whether they all provide new information. If not, the static dependence will lead to a rejection of one of the two parameters to the detriment of the more complex one to be calculated. The goal is to be parsimonious and to propose the least complex method. The Spearman's Rank Correlation Coefficient [67] is a statistical tool that examines the degree to which two datasets are correlated, even if the relationship between these two sets is not linear (unlike the Pearson one). This parameter is defined from the ratio between the covariance of two rank variables R(X) and R(Y) ($cov(R(X), R(Y))$) and the product of their standard deviation ($\sigma(R(X)x \ \sigma(R(Y))$). If the coefficient is greater than 0.75 (relationship in the same direction) or lower than −0.75 (relationship in the opposite direction), the correlation must be considered to be high. The task is then to use only descriptors with cross-correlations (in the Spearman sense) between −0.75 and 0.75. Figure 8 shows the Spearman correlation coefficient between the six descriptors of the kt series. A lot of values are outside the range of "statistical independence". To have only values between

−0.75 and 0.75, it is advisable to keep only three variables: mean, Hurst exponent (H), and forecastability coefficient (*F*). Figure 9 shows the median and quartiles (of the mean, H, and *F*) of the series. We note that *F* is relatively homogeneous on the dataset, and the two other parameters are more scattered mainly for the mean.

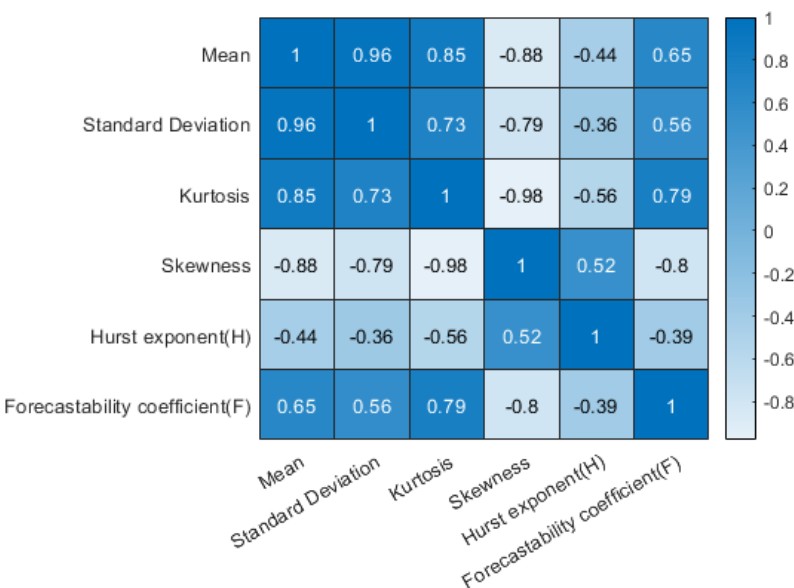

**Figure 8.** Descriptors correlations.

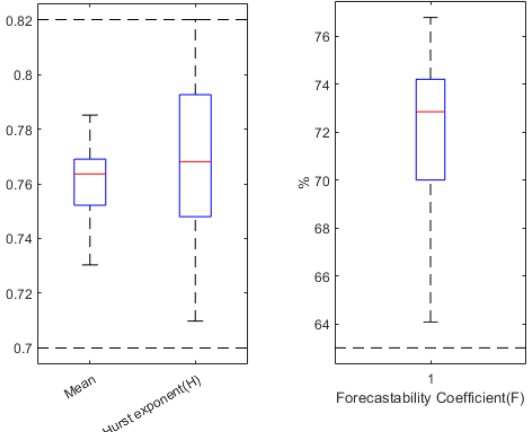

**Figure 9.** Boxplot: red line represents the median; blue lines represent the 3rd quartile (up) and 1st quartile (down); and black broken lines are the extremities—min and max values.

## 4. Köppen Climatic Classification and Clustering Methods

One of the simplest methods of geographic clustering based on the meteorological characteristics of the sites is the Köppen climatic classification. To facilitate the visualization of the results of this work, a Geographic Information System was implemented in QGIS software [68]. The various clusters are presented in the form of a map; several maps are drawn according to the clustering method applied. Thus, this makes it simpler to compare the various clusters with each other.

### 4.1. Köppen Climatic Classification

It consists of a worldwide natural climate classification [32] that identifies five main climate types, subdivided into a total of thirty classes, with a series of letters that indicate the behavior of temperatures and precipitations that characterize each climate and thus the kind of vegetation existing in them. Note that, a priori, the Köppen index (three

letters in Spain: first is climate type, second is rainfall regime, and third is temperature variations) has no direct relationship with the solar irradiance characteristic of the site. Considering that this original classification is relatively old (1981–2010 by the Instituto de Meteorología de España [69]) and that some climate modifications may have occurred, we decided to actualize the Köppen classification using the SIAR data and the monthly values of temperatures and precipitation between 2017 and 2020; a new Köppen classification was then elaborated as shown in Figure 10. As this Köppen classification is well-known worldwide, and even if the solar potential is not considered in this classification, it seems that it can be used in our study just for comparison purposes.

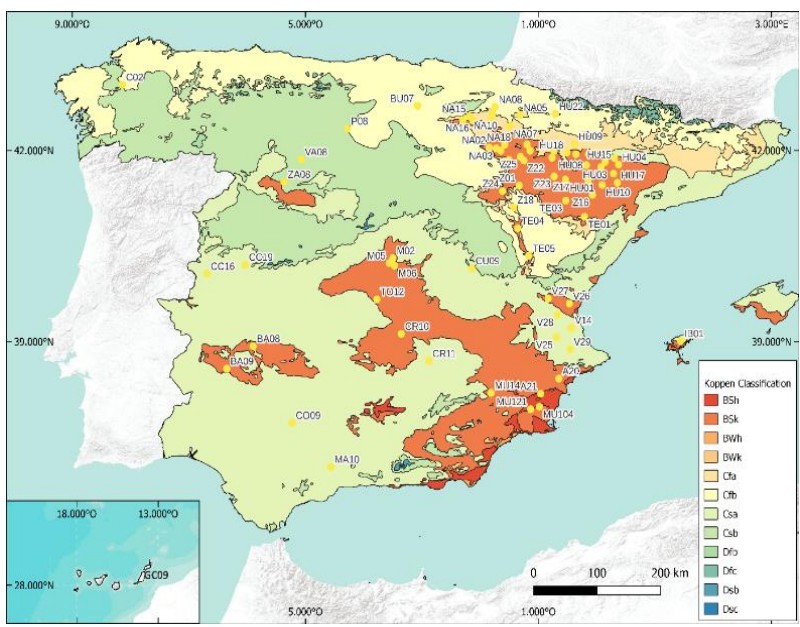

**Figure 10.** Köppen classification, updated from SIAR data for the years 2017–2020.

*4.2. Geographical Clustering Methods*

Clustering includes all techniques that involve the grouping of data points. Given a set of data points (characterized by several descriptive parameters), a clustering algorithm is used to classify each data point (i.e., meteorological stations) into a specific group. In theory, data points that are in the same group should have similar properties and/or features. In contrast, data points in different groups should have highly dissimilar properties and/or features. Machine learning, and particularly unsupervised learning, can be used for clustering. It is a common technique for data analysis used in many fields such as economics, medicine, market analysis, environmental-pattern analysis, crime analysis [19], business and socioeconomics, engineering, science, medicine, art, and entertainment, [16] among others. Time-series clustering can be classified into three categories [18,19,70]: (a) subsequence clustering is clustering on a set of subsequences of a time series that are extracted by using a sliding window, that is, clustering of segments from a single long time series [71]; (b) time-point clustering (time series segmentation) is a clustering of time points based on a combination of their temporal proximity of time points; and (c) whole time-series clustering is a clustering of a set of individual time series with respect to their similarity [72] (this paper is dedicated to this type of grouping).

4.2.1. The Available Tools

Applied to our geographical clustering problem, the methodology concerns a given set of data, D (SIAR data), from which we previously extracted indices about its static and dynamic behaviors. The problem of the whole clustering of time-series data is formally well defined in Reference [19] as follows: time-series clustering, given a dataset of n time-series data, $D = \{F_1, F_2, \ldots, F_n\}$, the process of unsupervised partitioning of $D$ into,

$C = \{C_1, C_2, \ldots, C_n\}$, in such a way that homogenous time series are grouped together based on a certain similarity measure, is called time-series clustering. Then $C_i$ is called a cluster, where $D = \cup_{i=1}^{k} C_i$ and $C_i \cap C_j = 0$ for $i \neq j$. A schematic of the main clustering techniques is shown in Figure 11.

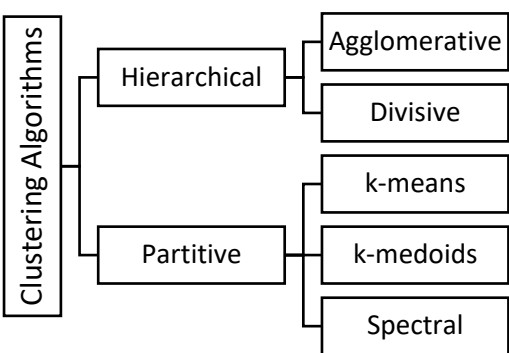

**Figure 11.** Main clustering techniques.

The hierarchical clustering algorithm is a method in which data are grouped based on the distance between each one and looking for the data within a cluster to be the most similar to each other. It can also be seen as a graphical representation where elements are nested in tree-like hierarchies [73]. Hierarchical clustering strategies can be either agglomerative (each observation starts in its own cluster) or divisive (all observations start as a single cluster). We decided to apply only the first one during simulations. Indeed, the purpose is to group the meteorological stations with the smallest distance between their dynamic and static characteristics.

The partitive clustering is different and allows to divide a dataset into k clusters by trying to minimize some specified error functions. In k-means and k-medoids cases, each data point belongs to a single group. Data points are assigned to a cluster such that the sum of the squared distance between the data points and the centroid of the cluster (means for k-means or median for k-medoid) is minimal [74,75]. Spectral clustering is a technique that reduces complex multidimensional datasets into clusters of similar data in rarer dimensions [76]. It is based on calculating the eigenvalues of the data-similarity matrix in order to reduce the dimensionality of the system.

Many details about clustering and the pitfalls to avoid are given in the valuable Hyndman papers [17]. In summary, k-means and k-medoids methods are the most commonly used clustering algorithm [77,78], with the number of clusters, k, specified by the user. Hierarchical clustering generates a nested hierarchy of similar groups of time series according to a pairwise distance matrix of the series [79]. Both of these clustering approaches, however, require that the length of each time series be identical due to the Euclidean distance calculation requirement and are unable to deal effectively with long-time series due to poor scalability. From a mathematical point of view now, k-means is defined from a set of observations $(x_1, x_2, \ldots, x_n)$. Each observation is a d-dimensional real vector; k-means clustering aims to partition the n observations into k ($\leq$n) sets S = $\{S_1, S_2, \ldots, S_k\}$ so as to minimize the within-cluster variance. Formally, the objective is to find the following ($\mu_i$ is the mean of points in $S_i$):

$$\underset{S}{\arg\min} \sum_{i=1}^{k} \sum_{x \in S_i} \|x - \mu_i\|2 \tag{5}$$

The hierarchical clustering is operated from sophisticated algorithms. In this technique, initially each data point is considered as an individual cluster. At each iteration, the similar clusters merge with other clusters until one cluster or k-clusters are formed. The basic algorithm is straightforward (agglomerative case):

- Compute the proximity matrix;
- Let each data point be a cluster;

-    Repeat: Merge the two closest clusters and update the proximity matrix;
-    Until only a single cluster remains.

As there is no consensus on which method to use, we chose to test all three in the next one. Table 2 offers a comparison of the main characteristics of the clustering methods used in this work.

**Table 2.** Comparison of the main characteristics of the clustering algorithms [80].

| Clustering Method | Basis of Algorithm | Input to Algorithm | Requires Specified Number of Clusters | Cluster Shapes Identified |
|---|---|---|---|---|
| Hierarchical | Distance between objects | Pairwise distances between observations | No | Arbitrarily shaped clusters, depending on the specified "Linkage" algorithm |
| k-Means k-Medoids | Distance between objects and centroids | Actual observations | Yes | Spheroidal clusters with equal diagonal covariance |
| Spectral | Graph representing connections between data points | Actual observations or similarity matrix | Yes, but the algorithm also provides a way to estimate the number of clusters | Arbitrarily shaped clusters |

A visible highlight in Table 2 concerns the fact that some clustering methods need to know the number of clusters to determine. How to choose the appropriate number of clusters is an open problem in the literature [81]. A pseudo-supervision can be used to define the ideal number of clusters (certainly the simplest way). For this reason, no algorithms were used to evaluate the optimal number of clusters. As the developed clustering was also compared with the Köppen classification (naive clustering method), it seemed convenient to take an equal number of clusters. Two cases were studied:

- According to the Spanish Köppen classification (BSh, BSk, Cfa, Cfb, and Csb), k = 5;
- According to the second letter (precipitation regime) of the Spanish Köppen classification (S, f, and s), k = 3. A strong link exists between GHI and precipitation: when one increases, the other decreases, and vice versa.

### 4.2.2. Clustering Methods Comparison

The notion of "objectively correct" does not exist in clustering: "clustering is in the eye of the beholder" [82]. The most appropriate clustering algorithm for a particular problem must often be chosen experimentally, unless there is a mathematical reason to prefer one clustering model over another. An algorithm designed for a specific model type will usually fail on a dataset containing a radically different model type. For example, k-means cannot find nonconvex clusters. According to the literature review, three important factors can be evaluated in clustering: (a) clustering tendency, which helps to evaluate whether the dataset we are working with has clustering tendency and not just uniformly distributed points [81,83]; (b) number of clusters k and (c) clustering quality. Concerning the last point, which is the most important for us, we can note several metrics which can be divided into internal and external measurements. The first one uses internal information to validate the clustering; that is, it evaluates how good the clustering structure is without the need for information from outside the algorithm itself [84,85]. Some of the clustering performance measures are the Davies–Bouldin Index, Silhouette Coefficient, Calinski–Harabasz Index, etc. The second one uses external information to validate the clustering and is independent of the clustering technique [86]. Some of the clustering-performance measures are the Rand Index, Jaccard Index, Fowlkes–Mallows scores, mutual-information-based scores, homogeneity, completeness, V-measure, etc. As it seems more relevant to us to use external measures, we have opted for the Rand Index method and the Jaccard Index:

- The Rand Index (RI) is a measure of the percentage of correct decisions made by the algorithm: RI = (TP + TN)/(TP + FP + FN + TN), where TP is the number of true

positives, TN is the number of true negatives, FP is the number of false positives, and FN is the number of false negatives;

- The Jaccard Index (JI) is a measure that quantifies the degree of similarity between two sets: JI = |A∩B| / |A∪B|, where A and B are two sets or clusters.

There is no real reason for this choice, except that their construction seemed better suited to the specificities of our study. The Rand and Jaccard Indices measure the degree of similarity between two sets. The range goes from 0% to 100%; if the partition is totally different, it is 0%, and if it is strictly identical, it is 100%. Note that the Jaccard Index ignores true negatives, while the Rand Index does not [87,88].

## 5. Results

While it is easy to show the results of clustering methods, it is not straightforward to validate it or to compare it objectively with another one. We therefore start by showing maps of Spain with the distribution of clusters according to the 76 stations studied, and then we compare each method by using appropriate metrics. All the results obtained by these clustering methods are detailed in Appendix A Table A2.

### 5.1. Case of 5 Clusters

As seen previously, we had to set a number, k, for this clustering study. We arbitrarily chose k = 5 because the Köppen classification states (empirically) that there are five different climates in Spain. The results of clustering are shown in Figures 12–15. As expected, the results related to k-means and k-medoids are equivalent. Indeed, when there is no or few outliers, these two methods are almost similar. Concerning the two other methods, there are small differences, but it is visually difficult to differentiate them. At this level, it is really difficult to distinguish between these four methods, and specific metrics will have to be used for more objectivity. Tables 3 and 4 are presented the Rand and Jaccard Indices. The Rand and Jaccard approaches for the cluster comparison are really similar in the study. If some differences are visible, the conclusions remain the same. There is no interest in using k-means, k-medoids, and spectral separately since the clusters resulting from these models are similar (similarity higher than 95% for Rand and 91% for Jaccard). The choice to use k-medoids, hierarchical, and Köppen for geographical clustering seems quite justified given this result and given that k-medoids is in theory more robust than k-means and spectral. Let us see now if, by decreasing the number of clusters (3), this conclusion remains the same.

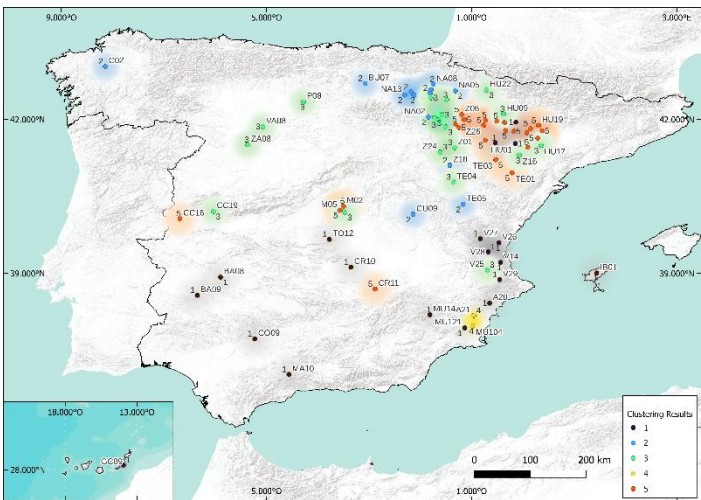

**Figure 12.** Using k-means for 5 clusters.



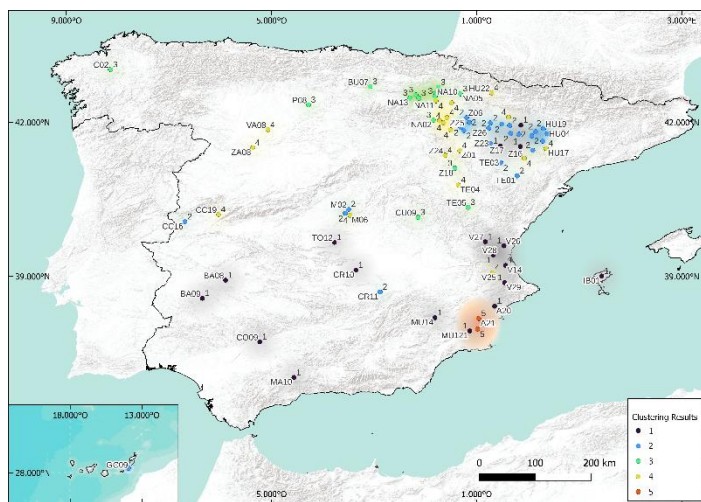

**Figure 13.** Using k-medoids for 5 clusters.

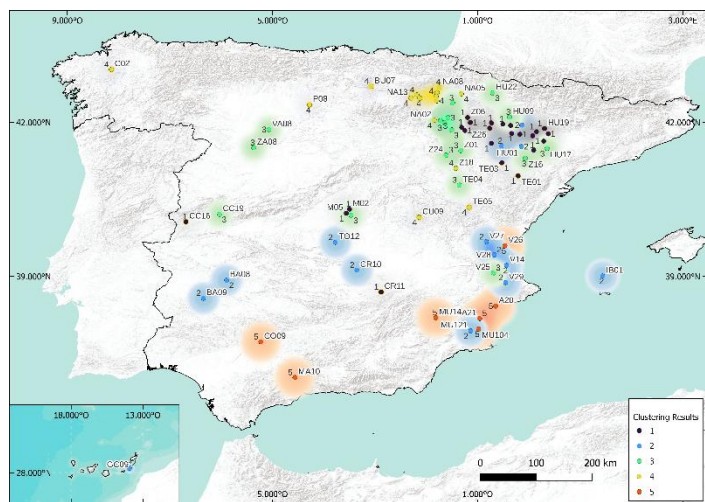

**Figure 14.** Using spectral for 5 clusters.

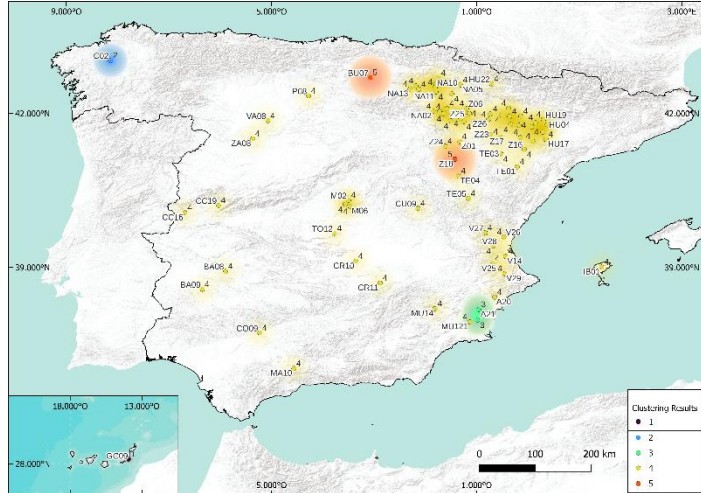

**Figure 15.** Using hierarchical for 5 clusters.

**Table 3.** Rand Index (RI) for 5 clusters.

|  | k-Means (%) | Spectral (%) | k-Medoids (%) | Hierarchical (%) | Köppen (%) |
|---|---|---|---|---|---|
| k-means | 100 | 95 | 99 | 35 | 53 |
| Spectral | 95 | 100 | 96 | 32 | 52 |
| k-medoids | 99 | 96 | 100 | 35 | 53 |
| Hierarchical | 35 | 32 | 35 | 100 | 51 |
| Köppen | 53 | 52 | 53 | 51 | 100 |

**Table 4.** Jaccard Index (JI) for 5 clusters.

|  | k-Means (%) | Spectral (%) | k-Medoids (%) | Hierarchical (%) | Köppen (%) |
|---|---|---|---|---|---|
| k-means | 100 | 91 | 99 | 37 | 49 |
| Spectral | 91 | 100 | 92 | 37 | 47 |
| k-medoids | 99 | 92 | 100 | 37 | 49 |
| Hierarchical | 37 | 37 | 37 | 100 | 64 |
| Köppen | 49 | 47 | 49 | 64 | 100 |

## 5.2. Case of 3 Clusters

In this study, the visual rendering of the cluster through the four methods mentioned above is available in Figures 16–19. The k-medoids method seems to stand out from the others, but the differences are still quite difficult to distinguish. Tables 5 and 6 present the Rand and Jaccard Indices.

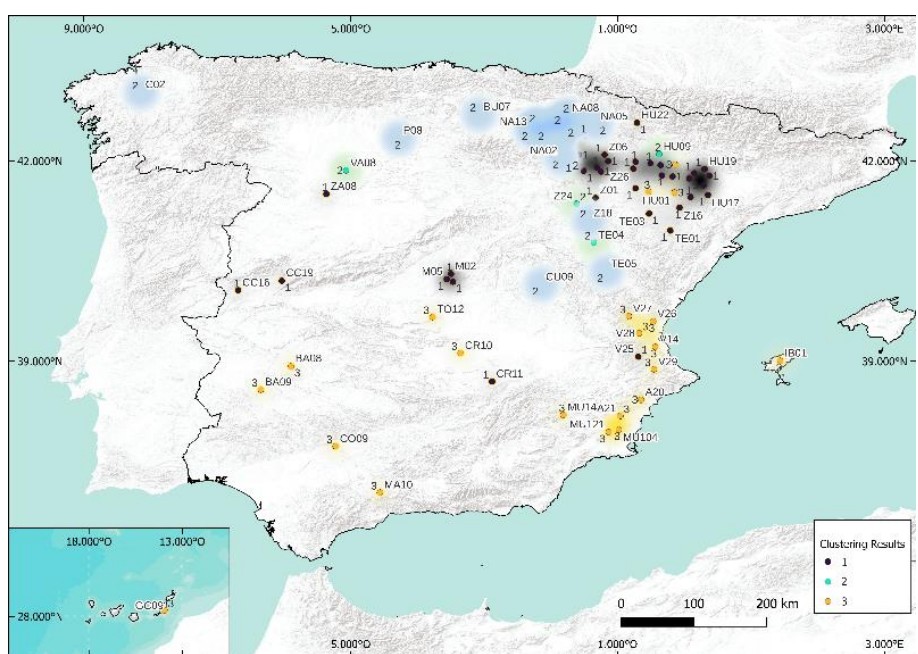

**Figure 16.** Using k-means for 3 clusters.

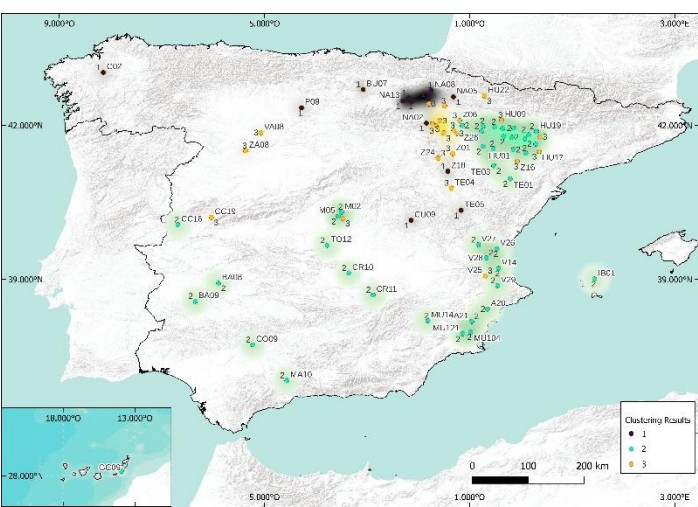

**Figure 17.** Using k-medoids for 3 clusters.

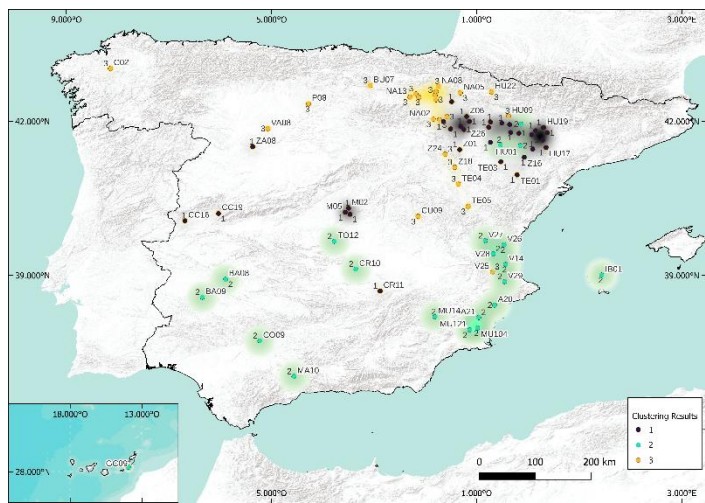

**Figure 18.** Using spectral for 3 clusters.

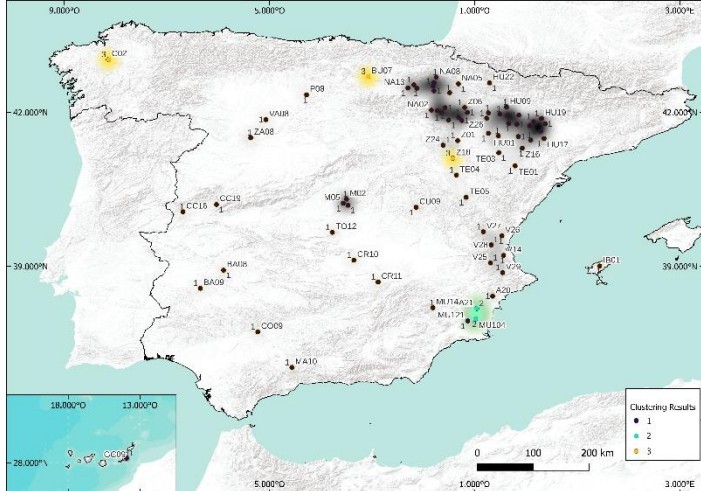

**Figure 19.** Using hierarchical for 3 clusters.

**Table 5.** Rand Index (RI) for 3 clusters.

|  | k-Means (%) | Spectral (%) | k-Medoids (%) | Hierarchical (%) | Köppen (%) |
|---|---|---|---|---|---|
| k-means | 100 | 95 | 70 | 41 | 51 |
| Spectral | 95 | 100 | 69 | 40 | 52 |
| k-medoids | 70 | 69 | 100 | 44 | 60 |
| Hierarchical | 41 | 40 | 44 | 100 | 53 |
| Köppen | 51 | 52 | 60 | 53 | 100 |

**Table 6.** Jaccard Index (JI) for 3 clusters.

|  | k-Means (%) | Spectral (%) | k-Medoids (%) | Hierarchical (%) | Köppen (%) |
|---|---|---|---|---|---|
| k-means | 100 | 96 | 67 | 53 | 58 |
| Spectral | 96 | 100 | 63 | 49 | 58 |
| k-medoids | 67 | 63 | 100 | 54 | 55 |
| Hierarchical | 53 | 49 | 54 | 100 | 62 |
| Köppen | 58 | 58 | 55 | 62 | 100 |

When analyzing the results obtained by k-means and spectral clustering algorithms, it can be said that they present similar results in more than 90% for classes three and five. The two tables (Rand and Jaccard) are similar; this suggests that the methodology is robust and the conclusions are stable. Contrary to case k = 5, the conclusions are slightly different here, and there would be a model to add: k-means or spectral. Because of its totally different foundation from that of k-medoid, our preference is for spectral. Thus, the methodologies to use (and to test) when studying predictions that require geographic clustering are, in this case, k-medoids, hierarchical, spectral, and Köppen.

## 6. Summary of Some Important Milestones of This Work

The methodology applied in this study problem is divided into two parts, as illustrated by two graphical schemes:

(a) The methodology used for analysis, correction, and validation of solar irradiance time series was presented in Figure 4. This part of the methodology was developed in Section 2.3, and the results obtained were presented in Section 2.4.

(b) The methodology used for characterization, clustering, and decision of the number of clusters from Köppen–Geiger classification is presented in Figure 20. This part of the methodology was developed in Section 3 (characterization); in Section 4, we presented the Köppen climatic classification and clustering methods; and finally, in Section 5, we presented the maps and results.

It should be noted that, in future work, if high-quality GHI data are available, the first part of the methodology (i.e., the one presented in Figure 4) may be unnecessary to implement. As a result, only the implementation of the second part of the methodology would be necessary in a future work (without the reduction of the number of characteristic variables from five to three (the three independent variables are sufficient). In conclusion, it can be evidenced that the second part of the proposed methodology is easy to implement.

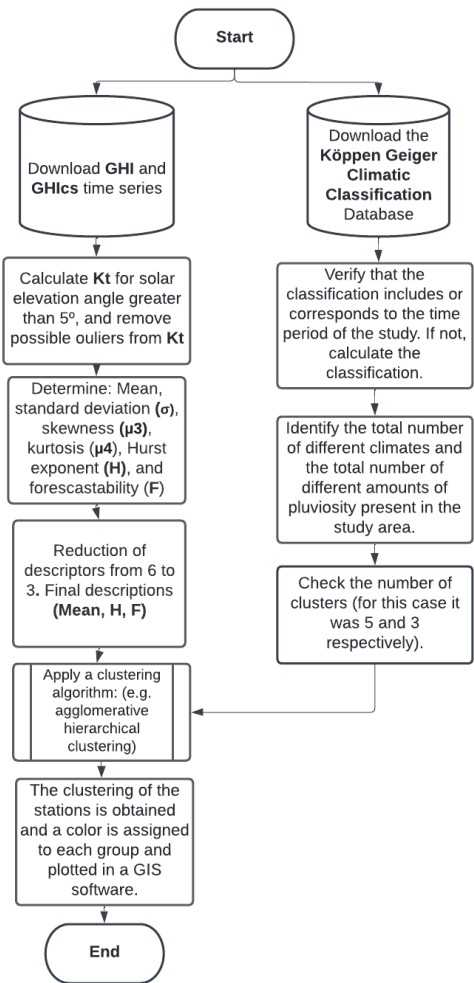

**Figure 20.** Various steps of the methodology.

## 7. Conclusions and Perspectives

The objective of this paper was to cluster several sites located in Spain according to solar radiation characteristics (quantitative and qualitative), with an objective to improve the forecasting of the solar irradiations. The novelties of this paper are not in the clustering methods, because the four methods used are well-known, but in the used parameters to qualify each meteorological station. Particularly, the use of the Hurst coefficient, and mainly the forecastability one, is original.

Each cluster groups the stations not according to the solar radiation potential only, as this has been already achieved for some sites over the World and is useful, as an example, for agricultural applications, but, and above all, it regroups the stations from the solar-radiation-variability point of view, i.e., from the qualitative viewpoint. It is essential to know that this variability in energy studies and this clustering will be useful for the elaboration of forecasting model based on time series and machine-learning methods.

The quality control of solar irradiation data provided by almost a hundred meteorological stations used in irrigation works was first realized during the study period 2017–2020. It allows us to be sure that the numerous data (more than 5 million GHI data) used in this work are relevant. After this quality control, 76 stations were then selected. In calculating the clearness index to know the clear-sky solar irradiance, the CAMs McClear model was used and previously verified; it appeared that its accuracy, when applied to our data, was high.

Firstly, each station was characterized, from a solar-radiation-variability point of view, by six parameters, including both static and dynamic ones: mean, standard deviation, skewness, Kurtosis, Hurst exponent, and forecastability coefficient. The dependence between these six parameters was then studied by using Spearman's Rank Correlation

Coefficient, and the results showed a high dependency between some of them; consequently, based on the parsimony principle and to avoid the redundancy of the information, only the mean, Hurst exponent, and forecastability coefficient were used to characterize each meteorological station.

From these descriptors, the stations were grouped by similarity (the shortest distance between descriptors), using the following algorithms: k-means, k-medoids, and spectral and hierarchical clustering to identify geographical areas with similar solar irradiation characteristics at a low time step (half an hour). The clusters were compared with each other and the well-known Köppen–Geiger climate classification.

The possible relationship between the Köppen classification and the number of clusters in which these time series can be grouped was analyzed. For both cases (five classes and three classes), the hierarchical clustering algorithm is the closest to the Köppen classification. However, it is probably not the most interesting to test, as it is a more complex method with the same (similar) results as a naive clustering method.

When evaluating the performance of the clustering algorithms, it can be concluded that there is no interest in simultaneously implementing k-means and spectral clustering since the results are similar in more than 90% for classes three and five.

The results shown in this study could be used to improve GHI prediction through the time-series formalism and machine learning in the following cases:

- To select to predictor considering that a method efficient on a meteorological site can be applied with the same efficiency of all the sites of the same cluster (i.e., with the same solar radiation variability or evolution);
- To make a forecast on a site where a GHI measurement device has just been installed. Clustering allows us to perform learning on data measured on one or several sites with similar characteristics and then to perform transfer learning to be able to make a forecast on the new site;
- To make GHI predictions on a site where few historical measurements are available. Clustering allows us to increase the number of training data (and thus the quality of the training by avoiding overfitting) by integrating data from other sites of the same cluster;
- To make prediction on a site where no measurement device is available. Clustering coupled with Kriging could allow for making a forecast from measurements obtained on nearby sites with similar meteorological characteristics;
- To improve a forecast by integrating during the learning process time-delayed data from other sites in the same cluster.

In all the previous cases, and based on the results shown previously, the authors propose to use k-means clustering or hierarchical clustering interchangeably (as many clusters as Köppen classes) based on mean, Hurst exponent, and forecastability computed from clear-sky index series (derived by CAMs estimation). However, for those who are less eager to obtain satisfactory results, the possibility of using the Köppen classification (recalculated for the time series study period) remains possible. It allows a classification that can be described as naive.

**Author Contributions:** This paper is the collaborative work of all authors. Conceptualization, L.G.-G. and C.V.; methodology, L.G.-G. and C.V.; software, L.G.-G.; validation, L.G.-G., C.V. and G.N.; investigation, L.G.-G.; resources, J.A.; data curation, L.G.-G.; writing—original draft preparation, L.G.-G.; writing—review and editing, C.V. and G.N.; supervision, C.V. and G.N. All authors have read and agreed to the published version of the manuscript.

**Funding:** This research received no external funding.

**Institutional Review Board Statement:** Not applicable.

**Informed Consent Statement:** Not applicable.

**Data Availability Statement:** The authors have not provided data associated with this article.

**Acknowledgments:** The authors would like to thank (i) the network of agroclimatic stations of the Spanish Ministry of Agriculture, Food, and Environment, specifically the agrometeorological stations network of SIAR for the information provided on solar irradiance (https://eportal.mapa. gob.es) (accessed on 2 June 2021; and (ii) the atmospheric monitoring service CAMs-McClear for the information on solar irradiance under clear sky conditions (http://www.soda-pro.com) (accessed on 10 March 2021).

**Conflicts of Interest:** The authors declare no conflict of interest.

## Appendix A

**Table A1.** Time-series characterization indices.

| Code | Station | Lat (°) | Long (°) | Altitude (m) | Köp Class 1981–2010 | Köp Class 2017–2020 | Mean | Standard Deviation | Kurtosis | Skewness | H | F |
|------|---------|---------|----------|--------------|---------------------|---------------------|------|--------------------|----------|----------|---|---|
| A20 | Agost | 38.422 | −0.650 | 941 | BSk | BSk | 0.81 | 0.44 | 3.39 | −1.15 | 0.73 | 71.91 |
| A21 | Orihuela | 38.180 | −0.959 | 96 | BSh | BSk | 0.82 | 0.44 | 3.87 | −1.30 | 0.73 | 74.25 |
| BA08 | Don Benito | 38.927 | −5.896 | 268 | BSk | BSk | 0.79 | 0.43 | 3.13 | −1.07 | 0.76 | 75.35 |
| BA09 | Vil. los Barros | 38.575 | −6.348 | 406 | BSk | BSk | 0.79 | 0.43 | 3.18 | −1.09 | 0.76 | 73.11 |
| BU07 | S. Gadea del Cid | 42.701 | −3.075 | 525 | Cfb | BSk | 0.67 | 0.37 | 1.79 | −0.36 | 0.75 | 51.96 |
| C02 | Boimorto | 43.032 | −8.141 | 429 | Csb | BSk | 0.61 | 0.37 | 1.68 | −0.20 | 0.76 | 50.28 |
| CC16 | Moraleja | 40.066 | −6.688 | 272 | Csa | BSk | 0.75 | 0.42 | 2.88 | −0.99 | 0.78 | 73.74 |
| CC19 | Zarza de Granadilla | 40.207 | −6.033 | 354 | Csa | BSk | 0.75 | 0.42 | 2.57 | −0.85 | 0.77 | 72.02 |
| CO09 | Palma del Río | 37.725 | −5.227 | 57 | Csa | BSk | 0.79 | 0.43 | 3.48 | −1.19 | 0.75 | 76.79 |
| CR10 | Manzanares | 39.122 | −3.354 | 645 | BSk | BSk | 0.80 | 0.44 | 3.15 | −1.12 | 0.76 | 74.59 |
| CR11 | Montiel | 38.696 | −2.881 | 887 | Csa | BSk | 0.79 | 0.44 | 2.86 | −1.01 | 0.76 | 72.42 |
| CU09 | Mariana | 40.152 | −2.142 | 941 | Csa | BSk | 0.73 | 0.42 | 2.16 | −0.69 | 0.77 | 67.25 |
| GC09 | Antigua | 28.320 | −13.942 | 72 | BWh | BSk | 0.81 | 0.43 | 3.13 | −1.01 | 0.66 | 73.83 |
| HU01 | Valfarta | 41.531 | −0.148 | 359 | BSk | BSk | 0.79 | 0.43 | 3.14 | −1.11 | 0.75 | 74.65 |
| HU02 | Zaidín | 41.636 | 0.289 | 182 | BSk | BSk | 0.78 | 0.43 | 2.89 | −1.04 | 0.77 | 76.32 |
| HU03 | Alcolea de Cinca | 41.740 | 0.073 | 225 | BSk | BSk | 0.80 | 0.44 | 2.86 | −1.05 | 0.77 | 75.72 |
| HU04 | Tamarite de Litera | 41.780 | 0.377 | 218 | BSk | BSk | 0.75 | 0.42 | 2.71 | −0.96 | 0.77 | 75.41 |
| HU05 | Lanaja | 41.786 | −0.338 | 361 | BSk | BSk | 0.79 | 0.43 | 2.93 | −1.01 | 0.76 | 74.02 |
| HU08 | Sariñena | 41.771 | −0.177 | 291 | BSk | BSk | 0.76 | 0.42 | 2.90 | −1.00 | 0.76 | 74.90 |
| HU09 | Huesca | 42.105 | −0.378 | 432 | Cfa | BSk | 0.74 | 0.42 | 2.42 | −0.78 | 0.77 | 74.17 |
| HU10 | Candasnos | 41.459 | 0.094 | 307 | BSk | BSk | 0.78 | 0.43 | 2.90 | −1.02 | 0.76 | 75.48 |
| HU11 | Grañén | 41.942 | −0.356 | 323 | BSk | BSk | 0.75 | 0.41 | 2.86 | −0.96 | 0.77 | 74.71 |
| HU12 | Huerto | 41.947 | −0.138 | 415 | BSk | BSk | 0.82 | 0.45 | 3.09 | −1.12 | 0.76 | 75.65 |
| Z06 | Ejea de los Caballeros | 42.097 | −1.196 | 316 | BSk | Cfa | 0.76 | 0.43 | 2.81 | −0.97 | 0.77 | 73.04 |
| HU13 | Gurrea de Gállego | 41.992 | −0.731 | 364 | BSk | BSk | 0.79 | 0.44 | 2.84 | −1.01 | 0.76 | 73.83 |
| HU15 | Alfántega | 41.821 | 0.148 | 249 | BSk | BSk | 0.77 | 0.43 | 2.86 | −1.02 | 0.77 | 75.33 |
| HU17 | Fraga | 41.494 | 0.354 | 98 | BSk | BSk | 0.75 | 0.42 | 2.52 | −0.88 | 0.78 | 73.75 |
| HU18 | Tardienta | 41.969 | −0.508 | 366 | BSk | BSk | 0.79 | 0.44 | 2.93 | −1.05 | 0.76 | 74.69 |
| HU19 | San Esteban de Litera | 41.882 | 0.304 | 316 | BSk | BSk | 0.81 | 0.45 | 2.82 | −1.04 | 0.77 | 76.01 |
| HU22 | Santa Cilia | 42.576 | −0.708 | 733 | Cfb | Csa | 0.76 | 0.43 | 2.47 | −0.87 | 0.77 | 71.23 |
| IB01 | Santa Eulalia del Río | 39.009 | 1.440 | 122 | Csa | BSk | 0.78 | 0.43 | 3.10 | −1.04 | 0.74 | 70.25 |
| M02 | Arganda del Rey | 40.310 | −3.498 | 531 | BSk | BSk | 0.80 | 0.44 | 2.91 | −1.07 | 0.76 | 73.43 |
| M05 | San Martín de la Vega | 40.233 | −3.560 | 516 | BSk | BSk | 0.78 | 0.43 | 2.83 | −1.01 | 0.77 | 73.65 |
| M06 | Chinchón | 40.192 | −3.469 | 534 | BSk | BSk | 0.78 | 0.43 | 2.63 | −0.94 | 0.77 | 72.70 |
| MA10 | Antequera | 37.034 | −4.563 | 457 | Csa | BSk | 0.79 | 0.43 | 3.39 | −1.18 | 0.75 | 73.46 |
| MU104 | Murcia | 37.977 | −0.984 | 128 | BSh | BSh | 0.85 | 0.46 | 3.84 | −1.33 | 0.73 | 74.75 |
| MU121 | Murcia | 37.939 | −1.135 | 54 | BSh | BSh | 0.83 | 0.45 | 3.21 | −1.13 | 0.73 | 73.97 |
| MU14 | Moratalla | 38.196 | −1.813 | 458 | BSk | BSk | 0.81 | 0.44 | 3.32 | −1.15 | 0.73 | 73.22 |
| NA02 | Fitero | 42.045 | −1.843 | 436 | BSk | Csa | 0.74 | 0.42 | 2.22 | −0.69 | 0.76 | 68.78 |
| NA03 | Cascante | 42.034 | −1.724 | 346 | BSk | BSk | 0.75 | 0.42 | 2.39 | −0.78 | 0.76 | 69.68 |
| NA04 | Ablitas | 41.996 | −1.645 | 338 | BSk | BSk | 0.77 | 0.43 | 2.49 | −0.88 | 0.76 | 70.05 |
| NA05 | Aibar/Oibar | 42.558 | −1.316 | 420 | Cfa | Csa | 0.75 | 0.42 | 2.22 | −0.71 | 0.78 | 69.71 |
| NA07 | Murillo el Fruto | 42.384 | −1.487 | 348 | Cfa | Csa | 0.75 | 0.42 | 2.57 | −0.89 | 0.77 | 70.52 |
| NA08 | Adiós | 42.686 | −1.747 | 443 | Cfb | Csa | 0.74 | 0.43 | 1.99 | −0.60 | 0.78 | 68.79 |
| NA09 | Artajona | 42.583 | −1.791 | 360 | Cfa | Cfa | 0.73 | 0.42 | 2.16 | −0.67 | 0.77 | 68.89 |
| NA10 | Miranda de Arga | 42.510 | −1.809 | 345 | Cfa | Csa | 0.73 | 0.41 | 2.17 | −0.65 | 0.78 | 69.05 |
| NA11 | Falces | 42.422 | −1.792 | 292 | Cfa | Csa | 0.75 | 0.42 | 2.30 | −0.74 | 0.77 | 69.31 |
| NA13 | Bargota | 42.477 | −2.299 | 382 | BSk | Cfa | 0.71 | 0.41 | 2.04 | −0.62 | 0.77 | 67.41 |
| NA15 | Arcos, Los | 42.539 | −2.185 | 421 | Cfa | Cfa | 0.72 | 0.41 | 2.03 | −0.59 | 0.77 | 67.10 |

**Table A1.** *Cont.*

| Code | Station | Lat (°) | Long (°) | Altitude (m) | Köp Class 1981–2010 | Köp Class 2017–2020 | Mean | Standard Deviation | Kurtosis | Skewness | H | F |
|------|---------|---------|----------|--------------|---------------------|---------------------|------|--------------------|----------|----------|---|---|
| NA16 | Sesma | 42.473 | −2.127 | 456 | Cfa | Csa | 0.73 | 0.42 | 2.08 | −0.62 | 0.77 | 67.75 |
| NA18 | Tudela | 42.093 | −1.577 | 243 | BSk | BSk | 0.76 | 0.43 | 2.50 | −0.83 | 0.77 | 71.60 |
| P08 | Lantadilla | 42.345 | −4.278 | 793 | Csb | Csb | 0.75 | 0.42 | 2.24 | −0.74 | 0.79 | 69.95 |
| TE01 | Calanda | 40.960 | −0.210 | 439 | BSk | BSk | 0.79 | 0.44 | 2.91 | −1.05 | 0.75 | 71.79 |
| TE03 | Híjar | 41.215 | −0.530 | 306 | BSk | BSk | 0.81 | 0.45 | 2.97 | −1.07 | 0.75 | 73.90 |
| TE04 | Monreal del Campo | 40.780 | −1.355 | 950 | BSk | BSk | 0.74 | 0.42 | 2.38 | −0.78 | 0.75 | 68.26 |
| TE05 | Teruel | 40.347 | −1.166 | 914 | BSk | BSk | 0.71 | 0.41 | 2.07 | −0.62 | 0.75 | 67.18 |
| TO12 | Mora | 39.664 | −3.773 | 735 | BSk | BSk | 0.81 | 0.44 | 3.15 | −1.13 | 0.76 | 72.49 |
| V14 | Algemesí | 39.216 | −0.436 | 19 | Csa | Csa | 0.79 | 0.44 | 3.10 | −1.06 | 0.74 | 73.92 |
| V25 | Bolbaite | 39.068 | −0.690 | 267 | Csa | Csa | 0.68 | 0.39 | 2.46 | −0.85 | 0.74 | 69.96 |
| V26 | Bétera | 39.598 | −0.468 | 97 | BSk | BSk | 0.78 | 0.43 | 3.48 | −1.18 | 0.74 | 73.82 |
| V27 | Chulilla | 39.676 | −0.832 | 378 | BSk | BSk | 0.80 | 0.44 | 3.18 | −1.13 | 0.73 | 73.14 |
| V28 | Godelleta | 39.421 | −0.677 | 270 | Csa | Csa | 0.77 | 0.43 | 3.20 | −1.11 | 0.74 | 71.87 |
| V29 | Bèlgida | 38.879 | −0.454 | 281 | Csa | Csa | 0.76 | 0.42 | 3.22 | −1.09 | 0.74 | 73.01 |
| VA08 | Medina de Rioseco | 41.860 | −5.071 | 727 | Csb | Csb | 0.77 | 0.43 | 2.44 | −0.81 | 0.78 | 71.24 |
| Z01 | Almonacid de la Sierra | 41.451 | −1.330 | 384 | BSk | BSk | 0.76 | 0.43 | 2.55 | −0.90 | 0.76 | 70.62 |
| Z14 | Borja | 41.854 | −1.508 | 378 | BSk | Csa | 0.77 | 0.43 | 2.50 | −0.87 | 0.76 | 70.33 |
| Z16 | Caspe | 41.303 | −0.071 | 175 | BSk | BSk | 0.75 | 0.42 | 2.67 | −0.86 | 0.77 | 74.69 |
| Z17 | Osera de Ebro | 41.544 | −0.537 | 251 | BSk | BSk | 0.80 | 0.44 | 3.04 | −1.11 | 0.76 | 74.63 |
| Z18 | Daroca | 41.107 | −1.425 | 748 | BSk | BSk | 0.68 | 0.40 | 1.77 | −0.40 | 0.74 | 64.08 |
| Z21 | Tauste | 41.999 | −1.143 | 353 | BSk | BSk | 0.78 | 0.43 | 2.88 | −1.01 | 0.77 | 72.31 |
| Z22 | Boquiñeni | 41.842 | −1.250 | 227 | BSk | BSk | 0.76 | 0.42 | 2.76 | −0.95 | 0.76 | 71.32 |
| Z23 | Pastriz | 41.593 | −0.731 | 182 | BSk | BSk | 0.81 | 0.45 | 2.90 | −1.03 | 0.76 | 74.29 |
| Z24 | Calatayud | 41.362 | −1.615 | 523 | BSk | BSk | 0.74 | 0.42 | 2.39 | −0.79 | 0.77 | 69.08 |
| Z25 | Tauste | 41.905 | −1.310 | 237 | BSk | BSk | 0.77 | 0.42 | 2.80 | −0.95 | 0.76 | 72.36 |
| Z26 | Zuera | 41.888 | −0.766 | 288 | BSk | BSk | 0.78 | 0.43 | 2.92 | −1.02 | 0.77 | 73.66 |
| ZA08 | Toro | 41.507 | −5.366 | 652 | Csb | Csa | 0.77 | 0.43 | 2.52 | −0.86 | 0.78 | 71.39 |

**Table A2.** Clustering by method and number of clusters.

| | | 5 Classes | | | | 3 Classes | | | |
|------|---------|---------|----------|-----------|--------------|---------|----------|-----------|--------------|
| Code | Station | k-Means | Spectral | k-Medoids | Hierarchical | k-Means | Spectral | k-Medoids | Hierarchical |
| A20 | Agost | 1 | 5 | 1 | 4 | 3 | 2 | 2 | 1 |
| A21 | Orihuela | 4 | 5 | 5 | 3 | 3 | 2 | 2 | 2 |
| BA08 | Don Benito | 1 | 2 | 1 | 4 | 3 | 2 | 2 | 1 |
| BA09 | Villafranca de los Barros | 1 | 2 | 1 | 4 | 3 | 2 | 2 | 1 |
| BU07 | Santa Gadea del Cid | 2 | 4 | 3 | 5 | 2 | 3 | 1 | 3 |
| C02 | Boimorto | 2 | 4 | 3 | 2 | 2 | 3 | 1 | 3 |
| CC16 | Moraleja | 5 | 1 | 2 | 4 | 1 | 1 | 2 | 1 |
| CC19 | Zarza de Granadilla | 3 | 3 | 4 | 4 | 1 | 1 | 3 | 1 |
| CO09 | Palma del Río | 1 | 5 | 1 | 4 | 3 | 2 | 2 | 1 |
| CR10 | Manzanares | 1 | 2 | 1 | 4 | 3 | 2 | 2 | 1 |
| CR11 | Montiel | 5 | 1 | 2 | 4 | 1 | 1 | 2 | 1 |
| CU09 | Mariana | 2 | 4 | 3 | 4 | 2 | 3 | 1 | 1 |
| GC09 | Antigua | 1 | 2 | 1 | 1 | 3 | 2 | 2 | 1 |
| HU01 | Valfarta | 1 | 2 | 1 | 4 | 3 | 2 | 2 | 1 |
| HU02 | Zaidín | 5 | 1 | 2 | 4 | 1 | 1 | 2 | 1 |
| HU03 | Alcolea de Cinca | 5 | 1 | 2 | 4 | 1 | 1 | 2 | 1 |
| HU04 | Tamarite de Litera | 5 | 1 | 2 | 4 | 1 | 1 | 3 | 1 |
| HU05 | Lanaja | 5 | 1 | 2 | 4 | 1 | 1 | 2 | 1 |
| HU08 | Sariñena | 5 | 1 | 2 | 4 | 1 | 1 | 2 | 1 |
| HU09 | Huesca | 3 | 3 | 4 | 4 | 2 | 3 | 3 | 1 |
| HU10 | Candasnos | 5 | 1 | 2 | 4 | 1 | 1 | 2 | 1 |
| HU11 | Grañén | 5 | 1 | 2 | 4 | 1 | 1 | 2 | 1 |
| HU12 | Huerto | 1 | 2 | 1 | 4 | 3 | 2 | 2 | 1 |
| Z06 | Ejea de los Caballeros | 5 | 1 | 2 | 4 | 1 | 1 | 3 | 1 |
| HU13 | Gurrea de Gállego | 5 | 1 | 2 | 4 | 1 | 1 | 2 | 1 |
| HU15 | Alfántega | 5 | 1 | 2 | 4 | 1 | 1 | 2 | 1 |
| HU17 | Fraga | 3 | 3 | 4 | 4 | 1 | 1 | 3 | 1 |
| HU18 | Tardienta | 5 | 1 | 2 | 4 | 1 | 1 | 2 | 1 |
| HU19 | San Esteban de Litera | 5 | 1 | 2 | 4 | 1 | 1 | 2 | 1 |
| HU22 | Santa Cilia | 3 | 3 | 4 | 4 | 1 | 3 | 3 | 1 |

**Table A2.** *Cont.*

| | | 5 Classes | | | | 3 Classes | | | |
|---|---|---|---|---|---|---|---|---|---|
| Code | Station | k-Means | Spectral | k-Medoids | Hierarchical | k-Means | Spectral | k-Medoids | Hierarchical |
| IB01 | Santa Eulalia del Río | 1 | 2 | 1 | 4 | 3 | 2 | 2 | 1 |
| M02 | Arganda del Rey | 5 | 1 | 2 | 4 | 1 | 1 | 2 | 1 |
| M05 | San Martín de la Vega | 5 | 1 | 2 | 4 | 1 | 1 | 2 | 1 |
| M06 | Chinchón | 3 | 3 | 4 | 4 | 1 | 1 | 3 | 1 |
| MA10 | Antequera | 1 | 5 | 1 | 4 | 3 | 2 | 2 | 1 |
| MU104 | Murcia | 4 | 5 | 5 | 3 | 3 | 2 | 2 | 2 |
| MU121 | Murcia | 1 | 2 | 1 | 4 | 3 | 2 | 2 | 1 |
| MU14 | Moratalla | 1 | 5 | 1 | 4 | 3 | 2 | 2 | 1 |
| NA02 | Fitero | 2 | 4 | 3 | 4 | 2 | 3 | 1 | 1 |
| NA03 | Cascante | 3 | 3 | 4 | 4 | 2 | 3 | 3 | 1 |
| NA04 | Ablitas | 3 | 3 | 4 | 4 | 1 | 1 | 3 | 1 |
| NA05 | Aibar/Oibar | 2 | 4 | 3 | 4 | 2 | 3 | 1 | 1 |
| NA07 | Murillo el Fruto | 3 | 3 | 4 | 4 | 1 | 1 | 3 | 1 |
| NA08 | Adiós | 2 | 4 | 3 | 4 | 2 | 3 | 1 | 1 |
| NA09 | Artajona | 2 | 4 | 3 | 4 | 2 | 3 | 1 | 1 |
| NA10 | Miranda de Arga | 2 | 4 | 3 | 4 | 2 | 3 | 1 | 1 |
| NA11 | Falces | 3 | 4 | 4 | 4 | 2 | 3 | 3 | 1 |
| NA13 | Bargota | 2 | 4 | 3 | 4 | 2 | 3 | 1 | 1 |
| NA15 | Arcos, Los | 2 | 4 | 3 | 4 | 2 | 3 | 1 | 1 |
| NA16 | Sesma | 2 | 4 | 3 | 4 | 2 | 3 | 1 | 1 |
| NA18 | Tudela | 3 | 3 | 4 | 4 | 1 | 3 | 3 | 1 |
| P08 | Lantadilla | 3 | 4 | 3 | 4 | 2 | 3 | 1 | 1 |
| TE01 | Calanda | 5 | 1 | 2 | 4 | 1 | 1 | 2 | 1 |
| TE03 | Híjar | 5 | 1 | 2 | 4 | 1 | 1 | 2 | 1 |
| TE04 | Monreal del Campo | 3 | 3 | 4 | 4 | 2 | 3 | 3 | 1 |
| TE05 | Teruel | 2 | 4 | 3 | 4 | 2 | 3 | 1 | 1 |
| TO12 | Mora | 1 | 2 | 1 | 4 | 3 | 2 | 2 | 1 |
| V14 | Algemesí | 1 | 2 | 1 | 4 | 3 | 2 | 2 | 1 |
| V25 | Bolbaite | 3 | 3 | 4 | 4 | 1 | 3 | 3 | 1 |
| V26 | Bétera | 1 | 5 | 1 | 4 | 3 | 2 | 2 | 1 |
| V27 | Chulilla | 1 | 2 | 1 | 4 | 3 | 2 | 2 | 1 |
| V28 | Godelleta | 1 | 2 | 1 | 4 | 3 | 2 | 2 | 1 |
| V29 | Bèlgida | 1 | 2 | 1 | 4 | 3 | 2 | 2 | 1 |
| VA08 | Medina de Rioseco | 3 | 3 | 4 | 4 | 2 | 3 | 3 | 1 |
| Z01 | Almonacid de la Sierra | 3 | 3 | 4 | 4 | 1 | 1 | 3 | 1 |
| Z14 | Borja | 3 | 3 | 4 | 4 | 1 | 1 | 3 | 1 |
| Z16 | Caspe | 3 | 3 | 4 | 4 | 1 | 1 | 3 | 1 |
| Z17 | Osera de Ebro | 1 | 2 | 1 | 4 | 3 | 2 | 2 | 1 |
| Z18 | Daroca | 2 | 4 | 3 | 5 | 2 | 3 | 1 | 3 |
| Z21 | Tauste | 5 | 1 | 2 | 4 | 1 | 1 | 2 | 1 |
| Z22 | Boquiñeni | 5 | 1 | 2 | 4 | 1 | 1 | 3 | 1 |
| Z23 | Pastriz | 5 | 1 | 2 | 4 | 1 | 1 | 2 | 1 |
| Z24 | Calatayud | 3 | 3 | 4 | 4 | 2 | 3 | 3 | 1 |
| Z25 | Tauste | 5 | 1 | 2 | 4 | 1 | 1 | 3 | 1 |
| Z26 | Zuera | 5 | 1 | 2 | 4 | 1 | 1 | 2 | 1 |
| ZA08 | Toro | 3 | 3 | 4 | 4 | 1 | 1 | 3 | 1 |

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
