# Peer review of "Evaluation and Comparison of Spatial Clustering for Solar Irradiance Time Series"

_applsci, doi:10.3390/app12178529_

Round 1

Reviewer 1 Report

The results look encouraging and motivating. But there are still some contents, which need be revised in order to meet the requirements of publish. A number of concerns listed as follows:

(1)   The abstract does not provide significant information and it should be revised to highlight the significant methodological contributions and conclusions.

(2)    In the introduction section, you should give the novelty and the contributions of your works.

(3)    The theoretical background of the proposed method is adequately detailed in the paper.

(4)   Literature survey is insufficient. You must add and review all significant similar works that have been done. For example, https://doi.org/10.3390/agriculture12060793; https://doi.org/10.1016/j.engappai.2022.105139ï¼›https://doi.org/10.1007/s10489-022-03719-6 ; https://doi.org/10.1109/JSTARS.2021.3059451 and so on.

(5)   Compared with the existing methods, the innovation of the proposed method needs more detailed description.

(6)   How about the computation complexity of the proposed method?

(7)   In Line 261, t (?????????????? 262 (?) =??? 263 ??(? + ∆?)) is how to obtain? The authors should provide a describing.

(8)   At Line 678, add the sections of the “Institutional Review Board Statement”, “Informed Consent Statement”, “Data Availability Statement”.

Author Response

Dear reviewer,

Thank you for your efficient review.

You will find the answer to your comments in the attached file.

With kind regards

Tha authors

Reviewer 2 Report

I find 3 major issues, related to the approach of the research.

First, there is not an actual application of the clustering results, or at least is not explored, or mentioned, not even a potential usage for the results of the study. In fact, it is stated that:

"The majority of the papers presented above proposes methods and
modes of validation crucial for the clustering of solar radiation", "v
arious techniques of clustering [17] will be applied in view to group the meteorological stationsby “solar radiation affinities” "

but I find the study incomplete.

The second major issue is related to the reproducibility of the experiments. To begin, the data is stated to be obtained from public databases:

"The solar irradiances for the present work were provided by the Sistema de Información Agroclimática para el Regadío (SIAR) system [18]"

Reference [18] does not led to the actual databases. It goes to a general website. In any case, the reference would be (if this was the site used) https://eportal.mapa.gob.es//websiar/Inicio.aspx

Moreover, this is utterly incomplete, since detailed information about the stations selected, which particular dates used, etc. is not stated.

And third:

It is stated that: "How to choose the appropriate number of clusters is an open problem in the literature [57]." As far as I know, it is not that open. You can choose to check several indexes, such as internal measures; Silhouette Width for compactness, connectivity, and
Dunn Index for considering both, or even more stability measures. Check some of the packages developed for this purposes, as the clValid from R. Usually several of these are checked in order to select the most adequate number of clusters, regarding several criteria

As the selection of the clusters is not validated properly, it kind of invalidates the subsequent study.

Other issues:

It is stated that:

"Most of the machine learning and AI methods can only be used withstationary time series [20]"

Which in my opinion and experience, may not be true. At least, is not what is defended in the reference 20, "Multilayer feedforward networks are universal approximators", which, at most, demonstrates that MLPs apprach functions, with no word about time series stated. Moreover, one may think that if the generation of time series data follows a function or a distribution, the paper defends the contrary of the stated in the manuscript.

In general the redaction of the document is poor. It requieres extensive corrections. For example, line 203, "or the same stations see Figure 3Figure.", line 199 "It appears clearly that the clear sky", etc. 

Even more weird phrase structures like line 188, " time-series of horizontal global", line 173 "used generally meteorological variables", and so on.

Figure 4 has low resolution, with small font in comparison with the text.

There are some statements in the document as: "The elimination or minimization of uncertainties in kt time series is very important [34-36]"

How much important is "very important", in terms of quantitative values? Using recursively along the document this type of statements is not scientifically adequate. In this particular case, 3 references are included. Does not any of them provide some value or some reasoning more than "very important"?

Section 4.2.1 includes very little information about the clustering methods used. At least, a brief mathematical explanation should be included.

Regarding the checking of the results for the clustering: "As it seems more relevant to us to use external measures, we have opted for, Rand Index method [63] andJaccard index." these should be mathematically described and properly referenced.

Author Response

Dear reviewer,

Thank you very much for your efficient review.

You can find the asnswers to your comments in the attached file.

With a kind regards,

The authors

Round 2

Reviewer 1 Report

According to the revised paper, I have appreciated the deep revision of the contents and the present form of this manuscript.  There is little content, which need be revised according to the comment of reviewer in order to meet the requirements of publish. A number of concerns listed as follows:

(1)  The abstract should be rewritten to reflect the significance of the proposed work.

(2) Please highlight your contributions in introduction.

(3) Conclusion: What are the advantages and disadvantages of this study compared to the existing studies in this area?

(4) More equations are necessary to explain the proposed method.

(5) The method in the context of the proposed work should be written in detail.

(6) To explore Comparative results with existing approaches/methods relating to the proposed work.

(7) In order to further highlight the introduction, some latest references should be added to the paper for improving the reviews part.

Author Response

The authors want to thanks the reviewers for their interesting and constructive remarks who helped us to improve the quality of this article.

In the revised word version of the paper, if you activate the “Track Changes” you will be able to see all the modifications realized. If you inactivate it, you will see the modification in red for reviewer 1.

Reviewer 1

According to the revised paper, I have appreciated the deep revision of the contents and the present form of this manuscript.  There is little content, which need be revised according to the comment of reviewer in order to meet the requirements of publish. A number of concerns listed as follows:

Dear reviewer, we hope that this new version of the document will be convincing and that it will address all the concerns you have listed.

  • The abstract should be rewritten to reflect the significance of the proposed work.

We changed some sentences in the abstract (Lines 14-15, 20-22, ; as we cannot write a too long abstract, we added several sentences in the introduction to show the reflectance of this work.

  • Please highlight your contributions in introduction.

We added the highlights at the end of the introduction (Lines 194-203).

  • Conclusion: What are the advantages and disadvantages of this study compared to the existing studies in this area?

There is not an advantage or a disadvantage; the objective of this clustering is specific because :

  • it groups meteorological stations by solar energy characteristics point of view what it is rather rare;
  • it groups the stations not only according to the solar radiation potential what has been already achieved for some sites over the World and is useful, as an example, for agricultural applications, but and above all, it regroups the stations from the solar radiation variability point of view i.e. from the qualitative viewpoint. It is essential to know this variability in energy studies and this clustering will be useful for the elaboration of forecasting model based on time series and machine learning methods.
  • Sentences were added in conclusion (Lines 723-734=
  • In the introduction, some sentences were written to explain the necessity to forecast the solar energy production for the development of this clean energy source, for an improvement of the performance of photovoltaic systems and at last, for making such intermittent production means more easily integrated and managed by the electrical dispatcher.

  • More equations are necessary to explain the proposed method.

As you suggested, we added equations to define Kurtosis, Skewness, Hurst exponent and correlation coefficient in the Spearman sense (Eq. 1 to 3 and lines 476-478. Eq; 5 was also added.

Some explanations were also added about the clustering methods (Lines 573-595).

  • The method in the context of the proposed work should be written in detail.

A new short paragraph was added just before the conclusion (paragraph 5 – From Line 716) with a summary of the steps realized In this study. A new figure (Figure 20) was also added.

  • To explore Comparative results with existing approaches/methods relating to the proposed work.

The proposed method is theoretically well developed and we believe that it is in line with the idea of proposing a quality clustering. Taking several static and dynamic parameters (never studied in solar radiation clustering) and using a knowledge model to determine the number of classes has never been proposed. Even if we did not find any clustering study based on SIAR data, it is likely that the approach presented in this paper is attractive, robust and relatively simple to implement. Some new references were added in the introduction to show the recent works in a similar topic but using different parameters for characterizing the stations.

  • In order to further highlight the introduction, some latest references should be added to the paper for improving the reviews part.

We added 26 new references :

  • [5] to [13] to explain why the solar irradiance forecasting is useful and what is the objective of this clustering from a solar radiation point of view.
  • [26] to [29] : to present four recent works on clustering applied to solar radiation.
  • [52] to [57] : to give more information on Kurtosis and skewness and to show that these parameters were used for characterizing other meteorological phenomena.
  • [60] to [62] : to give more information on Hurst coefficient and to show that this parameter was used for characterizing other meteorological phenomena.
  • [78] to [81] : to give more information on clustering methods

Reviewer 2 Report

Thank you for addressing my comments. The paper now its suitable for publication.

Still, let me remark one point, and it is that nowadays MLPs and other AI techniques does allow even real time, time series forecasting, it just require the adequate format of the data or the adequate topology or design of the method (you can find works on real time turbulent phase reconstuction and prediction, adaptive optics applications, etc.)

Author Response

The authors want to thanks the reviewers for their interesting and constructive remarks who helped us to improve the quality of this article.

In the revised word version of the paper, if you activate the “Track Changes” you will be able to see all the modifications realized. If you inactivate it, you will see the modification in red for reviewer 1.

Reviewer 2

Thank you for addressing my comments. The paper now its suitable for publication.

Still, let me remark one point, and it is that nowadays MLPs and other AI techniques does allow even real time, time series forecasting, it just require the adequate format of the data or the adequate topology or design of the method (you can find works on real time turbulent phase reconstuction and prediction, adaptive optics applications, etc.)

hank you for this comment and thank you for accepteing to review this work
